# Sensitivity and kinetics of signal transmission at the first visual synapse differentially impact visually-guided behavior

Ignacio Sarria[1], Johan Pahlberg[2], Yan Cao[1], Alexander V Kolesnikov[3], Vladimir J Kefalov[3], Alapakkam P Sampath[2], Kirill A Martemyanov[1]*

[1]Department of Neuroscience, The Scripps Research Institute, Jupiter, United States; [2]Jules Stein Eye Institute, Department of Ophthalmology, University of California, Los Angeles, Los Angeles, United States; [3]Department of Ophthalmology and Visual Sciences, Washington University in St.Louis, St. Louis, United States

**Abstract** In the retina, synaptic transmission between photoreceptors and downstream ON-bipolar neurons (ON-BCs) is mediated by a GPCR pathway, which plays an essential role in vision. However, the mechanisms that control signal transmission at this synapse and its relevance to behavior remain poorly understood. In this study we used a genetic system to titrate the rate of GPCR signaling in ON-BC dendrites by varying the concentration of key RGS proteins and measuring the impact on transmission of signal between photoreceptors and ON-BC neurons using electroretinography and single cell recordings. We found that sensitivity, onset timing, and the maximal amplitude of light-evoked responses in rod- and cone-driven ON-BCs are determined by different RGS concentrations. We further show that changes in RGS concentration differentially impact visually guided-behavior mediated by rod and cone ON pathways. These findings illustrate that neuronal circuit properties can be modulated by adjusting parameters of GPCR-based neurotransmission at individual synapses.

*For correspondence: kirill@scripps.edu

**Competing interests:** The authors declare that no competing interests exist.

## Introduction

Signaling via G protein coupled receptors (GPCR) mediates neurotransmitter actions and shapes many essential neuronal processes including vision, motor control, and cognition (*Wettschureck and Offermanns, 2005*). In addition to tremendous diversity in neurotransmitter receptors and the neuronal responses they modulate, GPCR pathways can be tuned in terms of response magnitude, sensitivity, and kinetics, oftentimes creating characteristic signature profiles specifically suited to meet the signaling demands of a neuronal circuit (*Marder, 2012*; *Nusbaum and Blitz, 2012*). However, the molecular mechanisms underlying the formation of specific signaling patterns, their relationship to synaptic processing, and ultimately to behavior remain poorly understood.

Signaling in GPCR pathways is initiated as the active receptor stimulates GDP/GTP exchange on G protein α subunits, triggering their dissociation from Gβγ subunits. These dissociated G proteins form active species that regulate the activity of second messenger enzymes, and either directly or indirectly modulate the opening of a number of ion channels (*Cabrera-Vera et al., 2003*; *McCudden et al., 2005*). It is now well recognized that the predominant role in controlling GPCR signaling belongs to the Regulator of G protein Signaling (RGS) proteins. RGS proteins stimulate GTP hydrolysis thus promoting subunit re-association and resulting in the termination of G protein signaling on a physiological timescale (*Ross and Wilkie, 2000*; *Hollinger and Hepler, 2002*). Studies with

**eLife digest** At the back of the eye, a structure called the retina contains several types of cell that convert light into the electrical signals that the brain interprets to produce vision. Cells called rods and cones detect the light, and then signal to other neurons in the retina that relay this information to the brain. Rods and cones are specialized to respond best to different visual features: cones detect color and can track rapid movement; whereas rods are more sensitive to low light levels and so enable night vision.

All rods and cones communicate with particular types of neuron called an 'ON bipolar cell': rods send their information to rod-specific ON bipolar cells and cones to cone ON-bipolar cells. To maintain the differences in how visual features are detected, the signals sent by the rod or cone cells need to be tuned separately. Previous studies showed that bipolar cells rely on the action of proteins called RGSs to control how information is passed from rods and cones to ON bipolar cells. However, how the RGS proteins produce their effects is not well understood, and neither is their impact on vision or behavior.

Sarria et al. used a genetic approach to create mice that progressively lost RGS proteins from their retina over the course of several weeks. Recording the nerve impulses produced by the bipolar cells as light shone on the retina revealed that RGS depletion affects these neurons in three ways: how sensitive they are to the signals sent by the rod and cone cells, how quickly they respond to a signal, and the size of the electrical response that they produce.

Sarria et al. then investigated how these changes affected the behavior of the mice. To test the response of the rod cells, the mice performed tasks in dim light. This revealed that it was only when the sensitivity of the bipolar cells decreased that the mice performed worse. However, in a task involving fast-moving objects that investigated the response of cone cells, only changes to the speed of the response affected vision. Therefore, the RGS protein has different effects on the signals from rod cells and cone cells. These findings will be useful for understanding how different light sensitive cells in the retina communicate their signals to extract important visual features, allowing us to both see well at night and track rapid changes in scenery on a bright sunny day.

knockout mice indicate that the elimination of RGS impacts key parameters of GPCR-driven signaling including sensitivity, temporal characteristics, and response amplitudes (*Chen et al., 2000*; *Heximer et al., 2003*; *Rahman et al., 2003*; *Fu et al., 2006*; *Posokhova et al., 2010*). These changes are often paralleled by distinct behavioral changes, for example, motor deficits, alterations in drug sensitivity, memory and reward behavior (*Traynor and Neubig, 2005*; *McCoy and Hepler, 2009*; *Xie and Martemyanov, 2011*).

Interestingly, the effect of RGS protein loss appears to be cell-type specific. For example, the elimination of the dominant RGS in rod photoreceptors profoundly slows deactivation kinetics while minimally affecting response amplitude or sensitivity (*Chen et al., 2000*; *Krispel et al., 2006*), whereas in hippocampal pyramidal neurons, moderate changes in response kinetics are accompanied by increased sensitivity (*Xie et al., 2010*). Altogether, these findings suggest that changes in relative RGS activity may filter the GPCR-generated responses, tuning response parameters to control signaling and ultimately behavior. Although intuitive, this hypothesis remains untested, as the constitutive elimination of RGS proteins typically results in an all-or-nothing impact on GPCR pathways, and compensatory changes during development further confound the interpretation of behavioral changes.

GPCR pathways play an especially critical role in enabling vision at low light levels. Single-photon responses generated by rod photoreceptors are transmitted to the downstream rod ON-bipolar cells (ON-BC). The high gain and rapid response of visual signals puts exquisite demands on both the timing and sensitivity of signal transmission at this synapse to enable rod reliable vision (*Okawa and Sampath, 2007*; *Pahlberg and Sampath, 2011*). The transient suppression of the glutamate release from photoexcited rods is sensed by the postsynaptic GPCR, mGluR6 (*Snellman et al., 2008*; *Morgans et al., 2010*). In darkness, mGluR6 activates a G protein, $G\alpha_o$, which maintains the TRPM1 effector channels closed. The opening of TRPM1 channels and resulting generation of the depolarizing response requires deactivation of $G\alpha_o$ (*Dhingra et al., 2000*; *Sampath and Rieke, 2004*).

Recent studies have indicated two RGS proteins, RGS7 and RGS11 play central role in this process (*Morgans et al., 2007*; *Mojumder et al., 2009*; *Cao et al., 2012*; *Shim et al., 2012*). In humans, loss of function mutations in mGluR6 or TRPM1 prevent the depolarizing activity of ON-BC and result in congenital stationary night blindness (*Dryja et al., 2005*; *Audo et al., 2009*; *van Genderen et al., 2009*). Similarly, knockout of mGluR6 (*Masu et al., 1995*), TRPM1 (*Morgans et al., 2009*; *Shen et al., 2009*; *Koike et al., 2010*), $G\alpha_o$ (*Dhingra et al., 2000*), or RGS7/RGS11(*Cao et al., 2012*; *Shim et al., 2012*) in mice also disrupts synaptic transmission between rods and ON-BC, completely abrogating responses of ON-BCs to light flashes. The total loss of signal transmission in these models has largely prevented dissection of the mechanisms that connect the fine-tuning of the GPCR signaling cascade to relevant physiological functions. Furthermore, cone photoreceptors are active under bright illumination and also signal through ON-BC neurons forming a distinct circuit. However, contributions of mGluR6 cascade elements to signal transmission at cone synapses, and their implications for vision, remain poorly understood.

In this study we used an inducible system to examine the effect of progressive RGS loss in mature and differentiated retinas on the ability of ON-BC neurons to process signals generated by rod and cone photoreceptors and its impact on visually-guided behavior. We describe how graded reductions in RGS concentration affect ON-BC light-evoked responses and relate these observations to the performance in visually-guided behavioral tasks. Our findings illustrate how changes in RGS activity may tune GPCR response properties to adapt circuit function to specific behavioral demands.

## Results

### Conditional knockout mouse model produces a gradual reduction in RGS concentration in mature, differentiated retinas

While elimination of either RGS7 or RGS11 alone does not substantially alter depolarizing activity of ON-BCs (*Mojumder et al., 2009*; *Chen et al., 2010*; *Zhang et al., 2010*), simultaneous loss of both proteins completely abolishes the responses of these cells to light flashes (*Cao et al., 2012*; *Shim et al., 2012*). In an effort to achieve an intermediate state of RGS expression we studied mice where elimination of one RGS is combined with the haploinsufficiency of the other (e.g., *Rgs11−/−: Rgs7+/−* or *Rgs11+/−: Rgs7−/−*). Evaluation of these mouse models by both electroretinography (ERG) and single cell recordings from rod ON-BCs revealed largely normal responses, that showed only minor delays in the onset time of the ERG b-wave (*Figure 1*; *Table 1*). Thus, normal depolarizing activity of ON-BCs is supported by a relatively low concentration of the RGS proteins.

To achieve a greater reduction in RGS concentration we developed a model for inducible RGS7 elimination. A conditional *Rgs7* allele (*Rgs7flx/flx*) was introduced in the *Rgs11−/−* background (cDKO) and the mice were further crossed with a ubiquitous driver line expressing inducible Cre-recombinase (cDKO:Cre+; *Figure 2A*). In the resulting model, RGS elimination can be induced postnatally by the oral administration of tamoxifen. We first characterized changes in RGS7 expression following tamoxifen administration in cDKO:Cre+ mice. Immunostaining of retinal cross-sections revealed a progressive loss of the RGS7-positive signal from the ON-BC synapses in the outer plexiform layer (*Figure 2B*). A reduction was evident in both the number of RGS7 puncta and staining intensity reaching minimal levels by day 35. During this timeframe, no changes in the postsynaptic concentration of mGluR6 were detected (*Figure 2B*). We further quantified the decline in RGS7 protein by quantitative Western blotting. Similar to the immunohistochemical analysis, we detected a progressive decrease in the RGS7-positive band (*Figure 2C*). Quantitative analysis found immunostaining and Western blotting data to be in good agreement, both showing monophasic decays in the RGS7 levels (*Figure 2D*). We further analyzed RGS7 concentration upon tamoxifen induction separately for synapses between rods and cones, and their respective ON-BCs (*Figure 2B,D*; *Figure 2—figure supplement 1*). Before tamoxifen administration (0 day), RGS7 levels in rod and cone ON-BC synapses were comparable, showing no more than a 5 ± 2% difference in staining intensity between the two. The time course of RGS7 loss induced by tamoxifen was also not different between rod and cone ON-BC synapses: half of RGS7 in rod ON-BC was found at 7.4 ± 1.5 days, vs 7.8 ± 1.3 days in cone ON-BCs. At day 35 following tamoxifen administration, RGS7 expression was reduced to 18 ± 3% in rod ON-BCs and to 17 ± 3% in cone ON-BC synapses. Importantly, the cytoarchitecture at the ON-BC dendritic tips of cDKO mice remained unaltered despite the RGS7 loss. Postsynaptic accumulations of both TRPM1 and mGluR6 and the alignment of pre- and post- synaptic

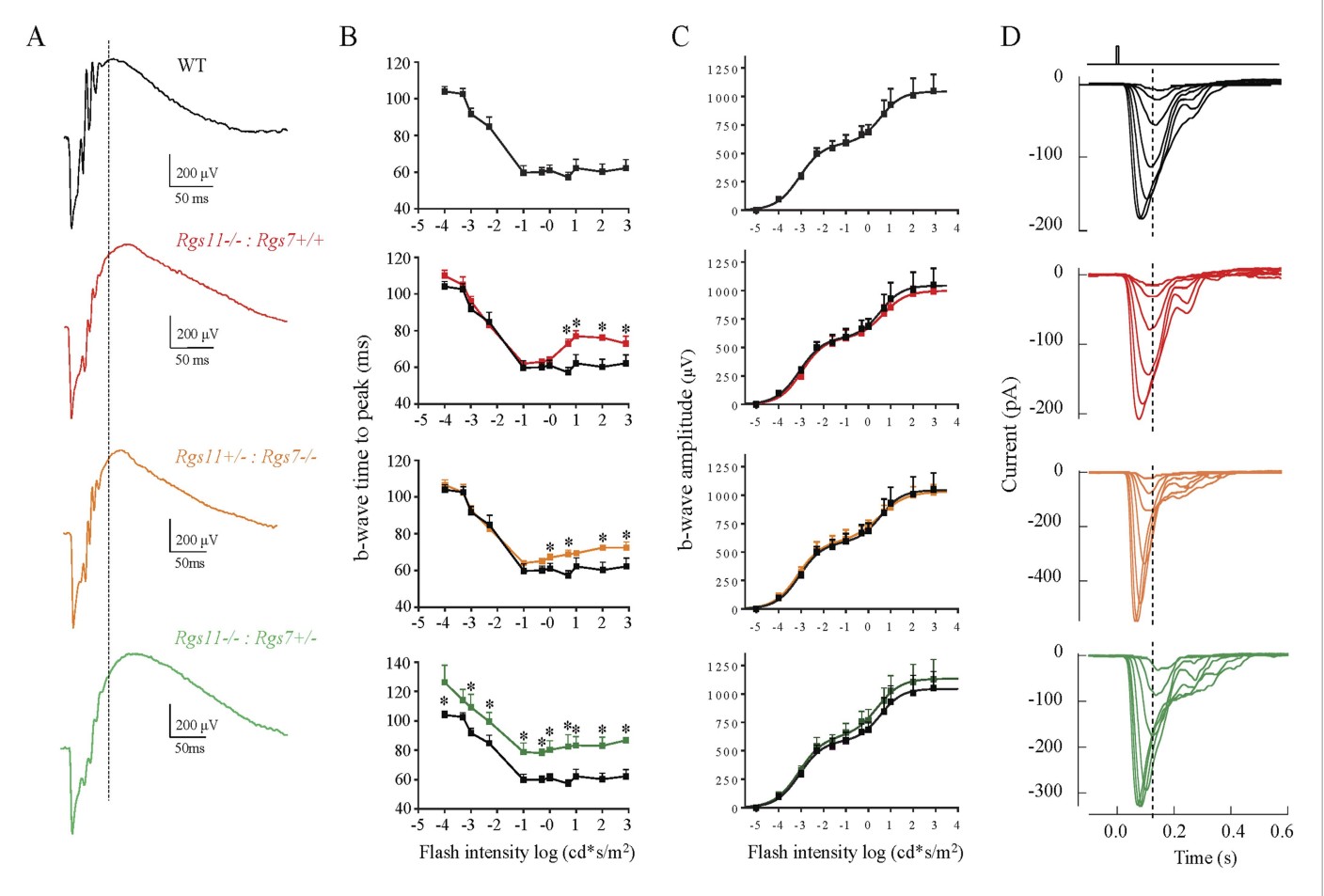

**Figure 1**. Effects of RGS haploinsufficiency on ON-BC properties. (**A**) Representative electroretinography (ERG) traces of RGS11 knockout (*Rgs11–/–*), RGS7 knockout (*Rgs7–/–*) alone or in combination with haploinsufficiency of the other (*Rgs11–/–*, *Rgs7+/–* and *Rgs11+/–*, *Rgs7–/–*). Dashed line shows position of the response peak recorded at the dimmest flash. (**B**) Quantification of ERG b-wave onset timing across increasing light intensities. Some genotypes showed minor delay in the onset of b-wave at moderate to high light intensities (>1 cd*s/m²). (**C**) Dependence of ERG b-wave amplitude on flash intensity (>1 cd*s/m² flashes [t-test, p > 0.5, n = 4–5]). Maximal amplitudes or sensitivities of b-waves were not affected in any of the genotypes. (**D**) Analysis of rod ON-BC light responses by single cell recordings. Responses of rod ON-BCs to flashes of light that varied in strength by factors of 2, from 0.5–23 R*/rod.

compartments were both preserved (*Figure 2E*; *Figure 2—figure supplement 2*). Prior to tamoxifen administration both cDKO:Cre- and cDKO:Cre+ mice were indistinguishable from the previously characterized *Rgs11–/–* background strain in both RGS7 expression and localization (*Figure 3A*). Furthermore, no changes in RGS7 concentration or its postsynaptic accumulation were detected upon

**Table 1**. ERG b-wave parameters extracted from fitting of scotopic and photopic phases of maximal b-wave amplitudes elicited by varying flash strengths

| Genotype | I₀.₅, scotopic (cd*s/m2) | Rₘₐₓ, (scotopic) (µV) | I₀.₅, photopic (cd*s/m²) | I₀.₅, photopic (µV) |
|---|---|---|---|---|
| WT | 0.0009 ± 0.0001 | 587 ± 17 | 3.5 ± 1.2 | 457 ± 30 |
| RGS11–/– | 0.001 ± 0.0001 | 580 ± 44 | 4.4 ± 1.4 | 419 ± 32 |
| RGS7–/– | 0.0008 ± 0.0001 | 615 ± 27 | 2.4 ± 1.1 | 510 ± 22 |
| RGS11–/– 7+/– | 0.0008 ± 0.0001 | 628 ± 51 | 2.5 ± 1.4 | 506 ± 25 |
| RGS11+/– 7–/– | 0.0007 ± 0.0001 | 593 ± 32 | 3.1 ± 1.0 | 439 ± 16 |

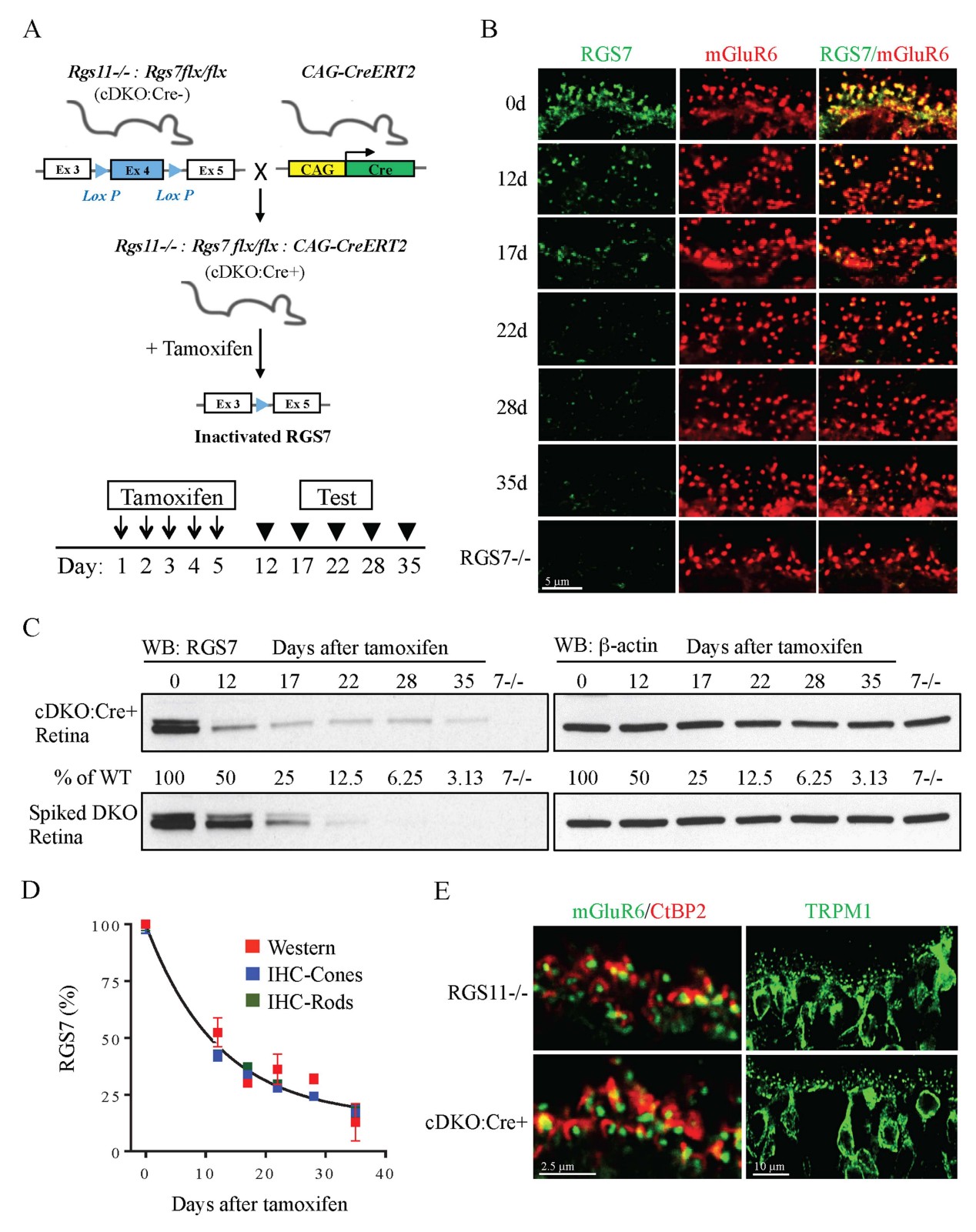

**Figure 2**. Progressive RGS7 loss in the ON-BC neurons of a conditional mouse knockout model (cDKO). (**A**) Schematic representation of a breeding strategy for the conditional inactivation of RGS7 on *Rgs11−/−* background. To induce recombination and RGS7 loss mice were administered tamoxifen for 5 days followed by testing. (**B**) Immunohistochemical analysis of RGS7 expression and localization in the outer plexiform layer of mouse retinas following

*Figure 2. continued on next page*

*Figure 2. Continued*

tamoxifen administration. Retina sections were stained with RGS7 (green) and mGluR6 (red) antibodies from cDKO:Cre+ mice at 12, 17, 22, 28, and 35 days after the start of tamoxifen administration. Retinas lacking RGS7 (*Rgs7*–/–) were used as a control. Two independent experiments yielding similar data were conducted. Note progressive loss of RGS7 immunoreactivity from postsynaptic puncta denoted by mGluR6. (**C**) Analysis of changes in RGS7 expression in total retina lysates by Western blotting. cDKO:Cre+ mice were treated with tamoxifen and retinas were collected at indicated time points. To generate retina lysates containing various RGS7 protein content, varying amounts of RGS11 knockout lysates were spiked into lysates isolated from RGS7 and RGS11 double knockout retinas (DKO). To determine the level of RGS7 reduction, a calibration curve was plotted from densities of varying amounts of RGS7 protein and used to determine the protein content in retina extracts obtained from tamoxifen-treated mice. Retinas from three separate animals were used in these experiments. (**D**) Quantification of RGS7 protein content at different time points following tamoxifen treatment. Values representing density of RGS7 band (red) and fluorescence intensity in mGluR6-positive puncta determined separately for rods (green) and cones (blue) were normalized to respective values observed before tamoxifen treatment (day 0). Averaged RGS7 percentages from western and IHC experiments are fitted with a single exponential decay function. Error bars are SEM values. (**E**) Intact synaptic morphology and retina cytoarchitecture at 28 days post-tamoxifen treatment. Retina cross-section from *Rgs11*–/– and cDKO:Cre+ mice were co-stained with postsynaptic marker mGluR6 and pre-synaptic marker CtBP2 and their direct apposition was noted. Immunostaining with TRPM1 (green) reveals intact morphology of ON-BC, their dendritic branching and distribution of postsynaptic puncta. *Figure 2—figure supplement 1*: assignment of rod vs cone synapses by immunohistochemistry. *Figure 2—figure supplement 2*: intact synaptic morphology and retina cytoarchitecture at 28 days post-tamoxifen treatment.

The following figure supplements are available for figure 2:

**Figure supplement 1**. Assignment of rod vs cone synapses by immunohistochemistry.

**Figure supplement 2**. Intact synaptic morphology and retina cytoarchitecture at 28 days post-tamoxifen treatment.

tamoxifen administration in cDKO littermates lacking Cre recombinase expression (*Figure 3A*). Consistent with these observations, ERG responses recorded in cDKO:Cre- and cDKO:Cre+ mice were indistinguishable from their *Rgs11*–/– controls and tamoxifen administration failed to alter ERG responses in cDKO:Cre- animals (*Figure 3B*). Thus, the observed effects are associated specifically with genomic editing of the *Rgs7* locus.

To obtain a more quantitative picture, we determined the absolute expression levels of RGS7 and RGS11 referencing them to the levels of mGluR6. Using recombinant protein standards and quantitative Western blotting, we found that RGS7, RGS11, and mGluR6 are present at $20 \pm 2$, $23 \pm 1$, $31 \pm 1$ fmol per 10 mg of total protein, respectively (*Figure 4*). Based on immunohistochemical analysis, RGS11 (*Cao et al., 2009*), RGS7 (*Cao et al., 2012*; *Figure 4—figure supplement 1*) and mGluR6 (*Masu et al., 1995*) are found exclusively, in the outer plexiform layer of the retina, indicating that the values that we obtained reflect protein content specifically at the ON-BC synapses. Considering that mGluR6 receptors form obligate dimers that function as a single unit, there is about threefold molar excess of RGS proteins (∼43 fmol) relative to mGluR6 dimer (∼15 fmol) in ON-BCs. In summary, these data illustrate that RGS7 loss beyond its stoichiometry with mGluR6 can be induced postnatally in a graded manner without affecting synaptic morphology or retinal cytoarchitechture.

## Progressive loss of RGS7 changes the kinetics and sensitivity of light-evoked responses in rod ON-BCs

To determine how progressive loss of RGS affects signal transmission in the rod pathway, we examined retinal responses to light in dark-adapted live anesthetized mice by ERG. In cDKO:Cre+ mice prior to tamoxifen administration, the delivery of very dim (scotopic) flashes, that activate rods but not cones, elicited a robust b-wave, whose generation requires the activity of rod ON-BCs (*Figure 5A*). These responses were indistinguishable from those observed in *Rgs11*–/– strain (*Figure 3B*). Administration of tamoxifen to cDKO:Cre+ mice dramatically changed these responses, progressively reducing their amplitude and slowing their time course (*Figure 5A*; *Table 2*). We next analyzed changes in response properties across a range of flash strengths. As flash strengths increase, the contribution of cones to the overall response becomes appreciable. However, rod- and cone-driven b-waves saturate at distinct light intensities leading to a characteristic biphasic shape of the intensity-response relationship (*Figure 5B*). Accordingly, the first phase of the curve is commonly considered to consist of relatively pure rod-driven component that can be extracted mathematically (*Figure 5C*; *Equation 1*). Analysis of this scotopic ERG b-wave showed marked time-dependent

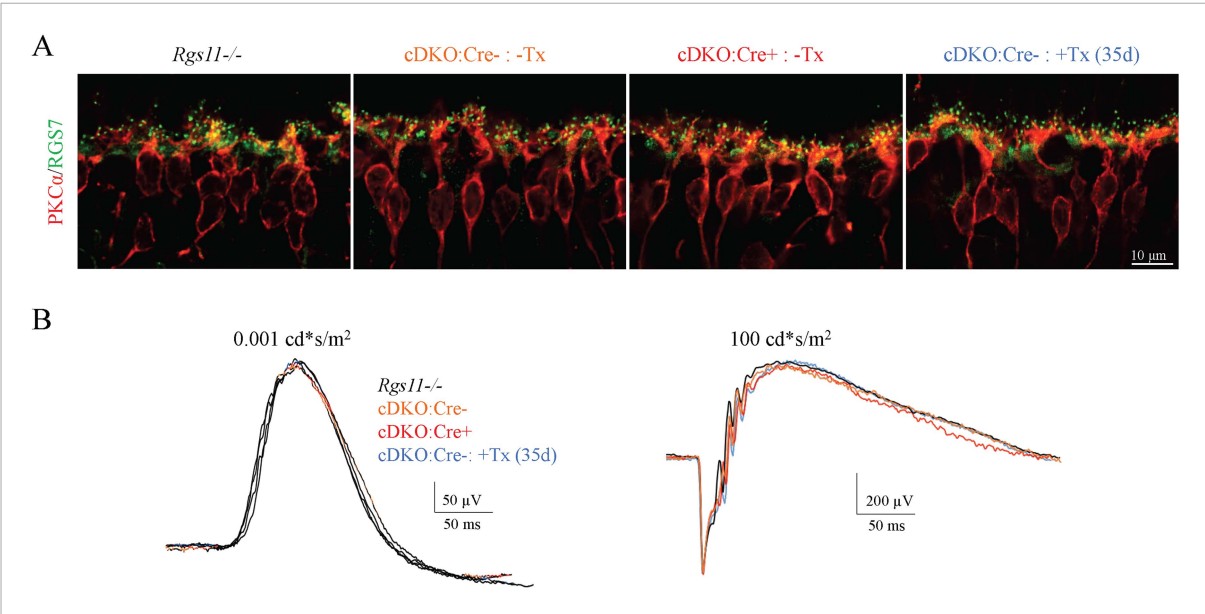

**Figure 3**. Expression of Cre recombinase without tamoxifen induction and tamoxifen administration without Cre expression do not alter RGS7 expression, localization and light response of mice. (**A**) Immunohistochemical analysis of RGS7 (green) and PKCα (red) expression and localization in the outer plexiform layer of *Rgs11–/–*, cDKO:Cre-, and cDKO:Cre+ mouse retinas. (**B**) Representative ERG traces elicited from scotopic (left) and photopic (right) flashes in mice described above show unaltered retina electrophysiological responses.

changes in both maximal response amplitude and sensitivity (*Table 2*). At 35 days following tamoxifen administration, the flash responses were barely distinguishable from the baseline seen upon constitutive elimination of both RGS7 and RGS11, with an approximately10-fold reduction in b-wave maximal amplitude and nearly a 100-fold decrease in its sensitivity. The responses further showed a time-dependent slowing in their onset, as evidenced by a progressive increase in b-waves time-to-peak and visible deceleration of the b-wave slopes (*Figure 5D*). Importantly, we observed no tamoxifen-induced changes in ERG b-wave characteristics in littermates lacking Cre recombinase (*Figure 3B*), indicating that these changes are caused specifically by the progressive loss in RGS7 expression.

To determine how the en masse activity of ON-BCs seen in ERGs is set by the responses of single cells, we recorded the light-evoked activity of rod ON-BCs in dark-adapted retinal slices (*Figure 6*). Whole-cell patch clamp recordings revealed some variability in ON-BCs responses suggesting that efficiency of tamoxifen-induced recombination varied on a cell-by-cell basis (*Table 3*). For example, we noticed that some cells were very susceptible to tamoxifen treatment and completely lost responsiveness to light, perhaps due to stochastic hyperactivation of Cre-recombinase leading to rapid elimination of RGS proteins. Because the proportion of unresponsive cells remained relatively stable (∼31–46%) across different days following tamoxifen administration, and because elimination of RGS proteins result in complete abrogation of ON-BC responses to flashes (*Cao et al., 2012*), these cells were excluded from the analysis. Analysis of the remaining light-sensitive cells revealed a progressive increase in time-to-peak and concomitant decrease in maximal amplitudes following tamoxifen administration (*Figure 6A*; *Table 3*). We next compared the average half-maximal flash strengths across all rod ON-BCs recorded on days 17, 22, 28, and 35 and found their progressive increase, reflecting a reduction in sensitivity. No changes in noise variance were detected, suggesting that reduction in RGS concentration also effectively decreases the signal-to-noise ratio (*Figure 6B*; *Table 3*). Interestingly, the sensitivity reduction determined from patch clamp recordings matched well that obtained by ERG, which summates the responses of all cells across the retina (*Figure 6C*). In summary, we observed that gradual titration of RGS levels affects kinetics, sensitivity, and amplitude of rod ON-BC light responses.

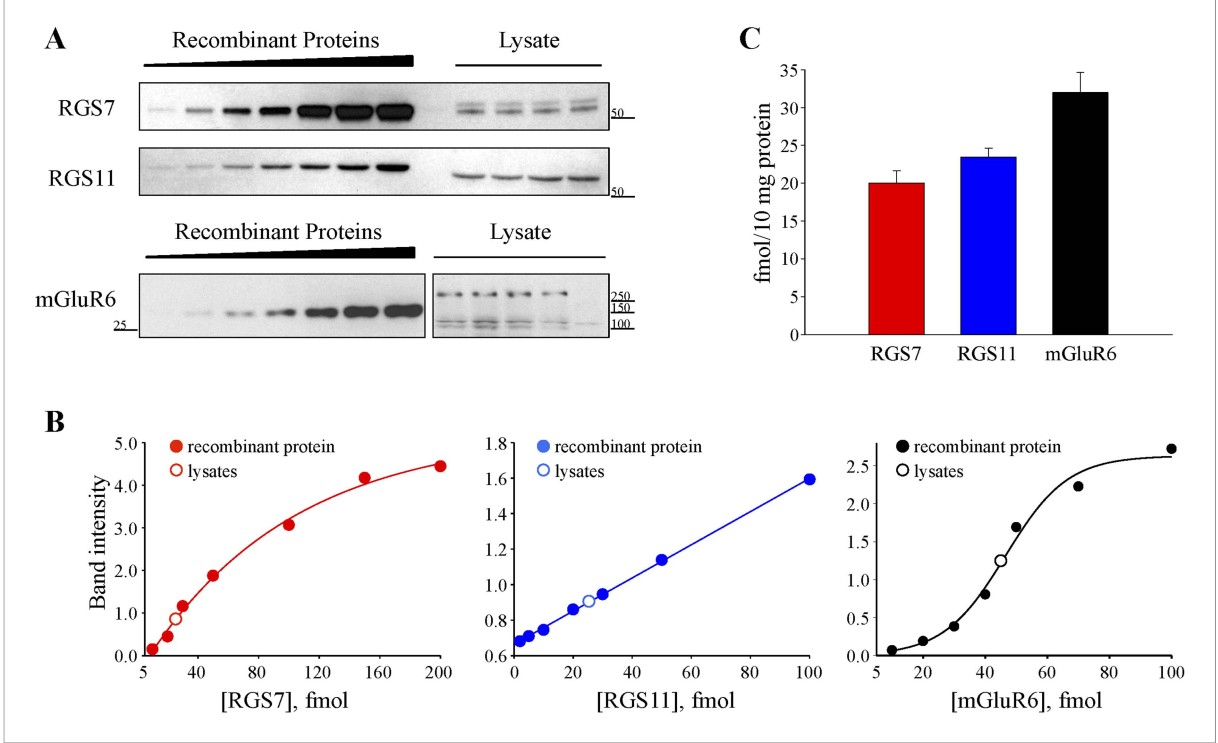

**Figure 4**. Quantification of RGS7, RGS11 and mGluR6 proteins in the retina. (**A**) Representative Western blots of retina lysates analyzed alongside with recombinant standards. (**B**) Quantification of protein content. A calibration curve was plotted from densities of recombinant protein standards (closed circles) and used to determine the protein content (open circles) in retina extracts obtained from four separate mice. (**C**) Comparison of absolute RGS7, RGS11 and mGluR6 protein levels in the retinas. The experiment was performed 3 times. Error bars indicate the SEM. *Figure 2—figure supplement 1*: distribution of RGS7 immunofluorescence in the retina.

The following figure supplement is available for figure 4:

**Figure supplement 1**. Distribution of RGS7 immunofluorescence in the retina.

## Decrease in RGS expression correlates with deterioration of the cone-to-cone ON-BC pathway function

Because ON-BCs making synapses with cone photoreceptors utilize a similar mGluR6-TRPM1 signaling pathway for generating depolarizing responses, we examined how changes in RGS concentration affected the light-evoked responses in cone ON-BCs (*Figure 7*, *Table 4*). Given low abundance of cones in murine retinas and the functional heterogeneity of cone ON-BC that they connect to, we relied exclusively on ERG for these studies. Prior to tamoxifen administration, bright (photopic) flashes produced a characteristic two-component ERG waveform in cDKO mice. In contrast, after tamoxifen administration we observed a progressive reduction in the b-wave amplitudes and kinetics without detectable changes in the a-wave (*Figure 7A*). We performed dose-response studies analyzing the dependence of photopic b-wave amplitude on flash intensity across all treatment groups. As in the case of rods, the component of the response that reflects cone contribution (*Figure 7B*) was derived mathematically from the biphasic response profile (*Figure 5C*). The resulting traces revealed changes in cone-generated responses (*Figure 7B*) with a progressive decrease in maximal response amplitude and sensitivity. To verify that recorded responses reflected specifically changes in cone ON-BC signaling, we performed ERGs when rod function was suppressed by exposure to background light (*Figure 7C*; *Figure 7—figure supplement 1*). Light-evoked responses collected on this rod-suppressing background showed a progressive decline in b-wave amplitude and kinetics, just as for dark-adapted conditions. Quantification of the parameters derived from the analysis of intensity-response relationships indicated that maximal amplitude, sensitivity, and

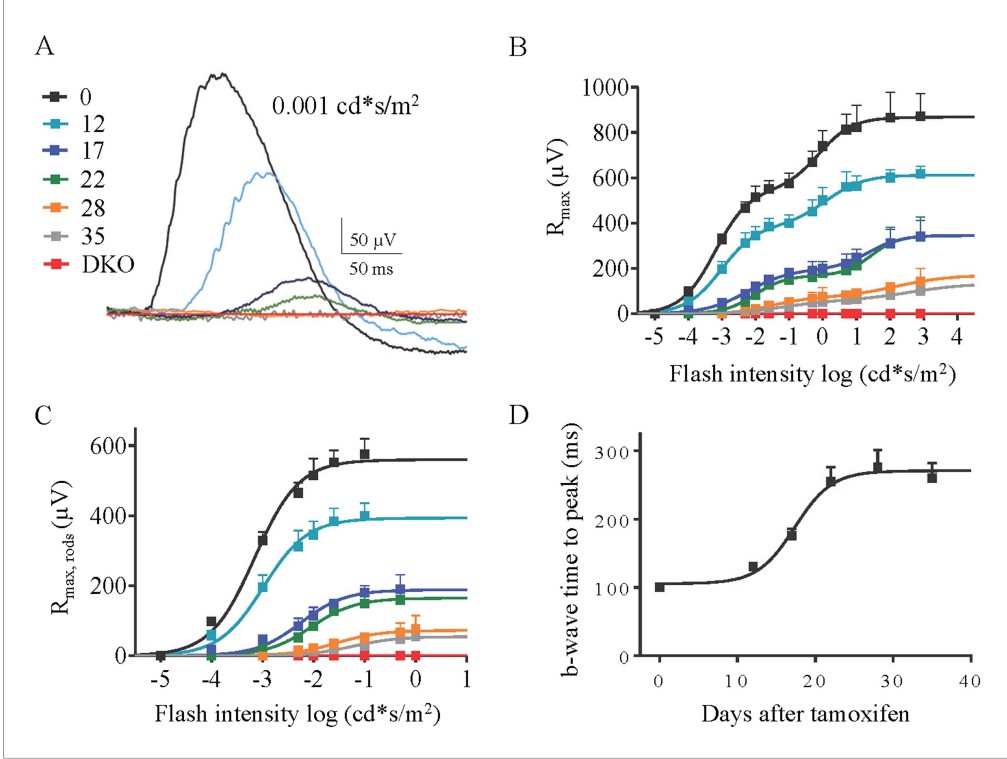

**Figure 5**. The effect of progressive RGS loss on rod ON-BC responses as revealed by scotopic ERG.
(**A**) Representative ERG responses elicited with dim flashes (0.001 cd*s/m²) from cDKO:Cre+ mice at 0, 12, 17, 22, 28, and 35 days post-tamoxifen administration. (**B**) Maximal ERG b-wave amplitudes plotted against their eliciting flash intensities. The ERG b-waves exhibit biphasic saturation pattern, with first one around 0.05 cd*s/m², corresponding to activity of rod ON-BC, and a second one around 100 cd*s/m², corresponding to activity of cone ON-BC. (**C**) Rod component of the ERG response extracted from fitting the first phase of the response in panel **B**. (**D**) Analysis of the changes in the onset of ERG b-wave elicited by half-saturating flashes. Time to peak values are plotted as a function of days after tamoxifen treatment. Errors bars are SEM.

onset timing of the response similarly declined with time after tamoxifen administration (*Figure 7D,E*). Together, these results indicate that down-regulation of RGS proteins in the cone synapses lead to changes in cone ON-BC light-evoked responses, that are similar to changes observed in rod synapses.

## Loss of RGS differentially affects visually-guided behavior under scotopic and photopic conditions

To assess how changing responses in rod ON-BCs affects dim light vision we developed a behavioral water maze assay. The assay measures the ability of mice to locate an escape platform under varying

**Table 2**. Effect of tamoxifen treatment on scotopic ERG b-wave parameters

| Days post tamoxifen | $I_{0.5}$, scotopic (cd*s/m²) | $R_{max}$, scotopic (μV) | Time to peak (ms) at $I_{0.5}$ |
|---|---|---|---|
| 0 | 0.0007 ± 0.0001 | 560 ± 10 | 100 ± 6 |
| 12 | 0.001 ± 0.0002 | 390 ± 10 | 130 ± 5 |
| 17 | 0.005 ± 0.0001 | 190 ± 8 | 180 ± 10 |
| 22 | 0.008 ± 0.0003 | 160 ± 5 | 260 ± 21 |
| 28 | 0.03 ± 0.001 | 70 ± 3 | 280 ± 25 |
| 35 | 0.06 ± 0.006 | 50 ± 4 | 260 ± 22 |
| DKO | N.A | 0 | N.A |

Values are mean + SEM, n = 5 mice for all timepoints.

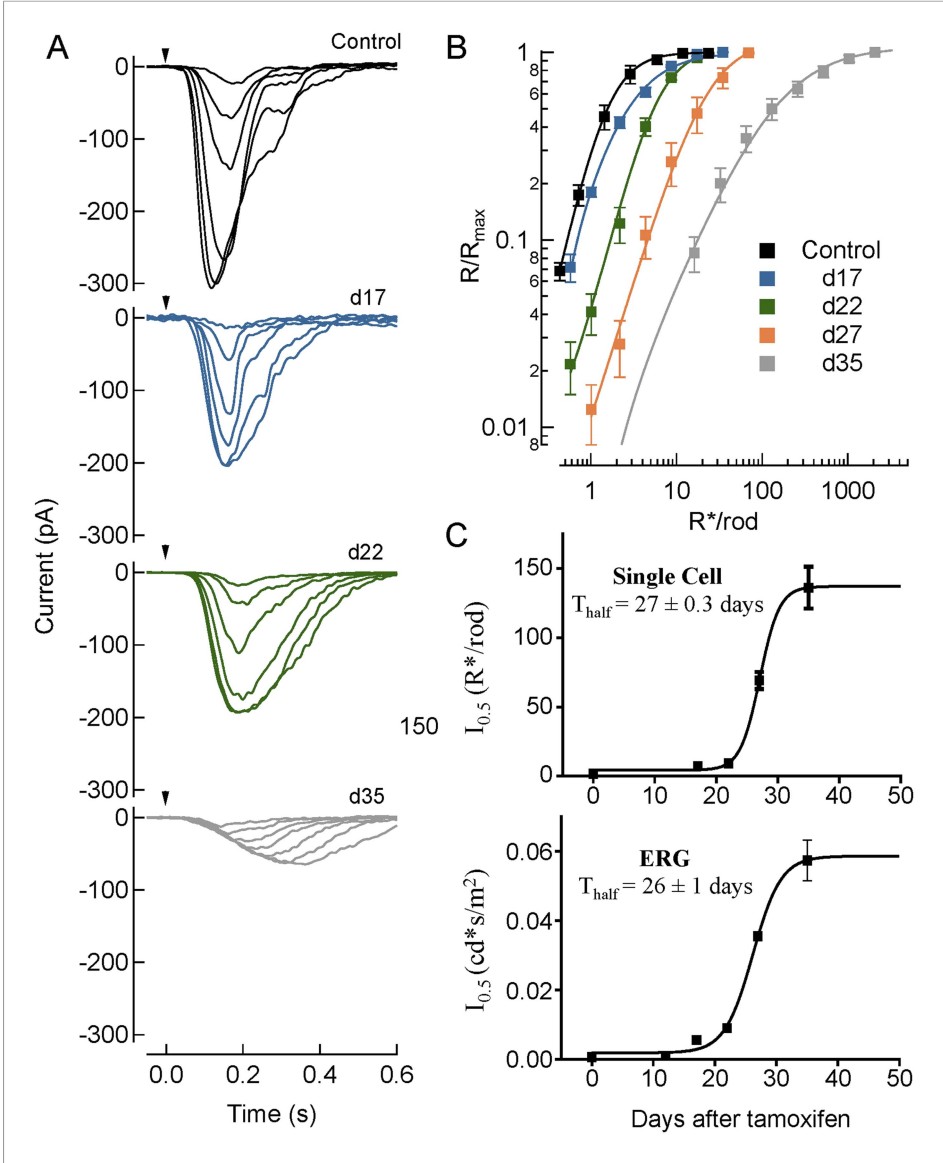

**Figure 6**. Single cell recordings from rod ON-BC recapitulate key changes in average light-evoked responses measured by ERG. (**A**) Responses of rod ON-BCs to flashes of increasing strength. Shown are representative response families for 10 ms flashes (arrow) measured at an increasing number of days following tamoxifen administration. Flash strengths yielded 0.70, 1.4, 2.9, 5.7, 11, and 23 R*/rod for control (*Rgs7*+/+:*Rgs11*−/−), 0.58, 1.0, 2.2, 4.3, 8.6 and 17 R*/rod for day 17; 1.0, 2.2, 4.3, 8.7, 17, and 35 R*/rod for day 22, and 16, 32, 65, 130, 260, 520, and 1000 R*/rod for day 35. Note with time following tamoxifen administration the progressive slowing and reduction in maximum amplitude of the light-evoked response (see *Table 3*). (**B**) Average normalized response-intensity relationships for all cells recorded for control (day 0) as well as days 17, 22, 28, and 35 following tamoxifen administration. Data were sampled at 1 kHz and filtered at 50 Hz. (**C**) Comparison of half-saturation responses from ON-BC measured by single-cell recordings (top) (n = 4–11) and ERG (bottom) (n = 5) from cDKO:Cre+ mice at different days after tamoxifen administration. Data points plotted are half-maximal values from each tamoxifen time point fitted by sigmoidal dose-response (variable slope) equation with the least squares method. The time required for tamoxifen to reduce the sensitivity of ON-BC by half is not different when obtained by ERG (28 ± 1 days) vs single cell patch-clamp (27 ± 0.3 days). Errors bars are SEM.

luminance levels. Under photopic conditions, trained wild-type mice reached the visible platform in ~10 s (*Figure 8A*). Video rate analysis of tracks indicated that under these conditions mice swim directly to the visible platform taking the shortest path (*Figure 8A*). In contrast, when the platform was

**Table 3.** Parameters of rod ON-BC light responses by single cell recordings

| Days post tamoxifen | $I_{0.5}$, (R*/rod)* | $R_{max}$ (pA) | Noise Variance (pA)$^2$ | Unresponsive† (μV) |
|---|---|---|---|---|
| 0 | 1.6 ± 0.10 (4) | 340 ± 33 | 1.2 ± 0.2 | 0/4 |
| 12 | 4.8 ± 1.6 (9) | 200 ± 52 | 2.1 ± 0.4 | 4/13 |
| 22 | 6.4 ± 0.10 (8) | 200 ± 31 | 1.6 ± 0.2 | 5/13 |
| 28 | 69 ± 31 (6) | 130 ± 45 | 1.3 ± 0.1 | 5/11 |
| 35 | 160 ± 34 (13) | 68 ± 14 | 2.1 ± 0.4 | 11/24 |

*Mean + SEM, number of cells is indicated in parenthesis.
†Unresponsive cells were those where a flash did not evoke a response.

hidden mice randomly swam and appeared to encounter the platform by chance, increasing the time to escape by nearly fivefold. Thus, mouse vision in this test can be evaluated by measuring the time to escape on the platform.

To validate the test, we evaluated the performance of the constitutive RGS7/11 double knockouts (DKO), which completely lack an ERG b-wave and thus are considered night blind. Under photopic conditions, the escape latency of DKO mice with the visible platform was no different from that of wild-type animals, indicating that they see the platform normally (**Figure 8A**). However, when the maze luminance was decreased to the scotopic range, the performance of DKO mice dropped sharply and their escape latencies became statistically equivalent to random encounters with the hidden platform, indicating a complete loss of scotopic vision (**Figure 8A**). Thus, RGS expression in rod ON-BCs is absolutely required for high sensitivity scotopic vision.

We next tested the performance of the cDKO mice under scotopic conditions. Before tamoxifen administration escape times of cDKO:Cre+ mice were substantially shorter than in DKO and no different from these of cDKO:Cre- littermates or WT mice, indicating that their scotopic vision remained intact (**Figure 8B**). This performance did not decline up to 22 days after the beginning of tamoxifen treatment. However, a marked increase in escape time was observed in cDKO:Cre+ mice at 28 days, and by 35 days they displayed similar difficulty in visually locating the platform as the constitutive DKO animals, indicating that they developed night blindness (**Figure 8B**). Administration of tamoxifen to RGS7 haploinsufficient or *Rgs7*−/− mice did not alter the performance compared to wild-type animals at any light levels (**Figure 8C**).

To understand better the relevance of RGS function in ON-BCs to behavior, we also assessed visual function of mice in a virtual environment using the optokinetic reflex (OKR). The optomotor response test is based on the ability of mice to respond reflexively to computer-generated rotating sine-wave gratings, which form a virtual cylinder around them (**Prusky et al., 2004**). Previous studies have established an essential contribution of ON-pathway to visual contrast sensitivity under both scotopic and photopic conditions reflecting the critical role of both rod and cone pathways to this behavior (**Schiller et al., 1986**; **Iwakabe et al., 1997**; **Kolesnikov et al., 2011**). To assess scotopic contrast sensitivity change in cDKO mice we used the optimal mouse scotopic temporal frequency of 0.75 Hz in combination with a moderate stimuli speed of 7.5 deg/s (**Umino et al., 2008**) at which the contrast sensitivity is close to maximal (**Figure 9A**). Before tamoxifen administration, both genotypes were indistinguishable in their ability to visualize rotating gratings at their contrast threshold. Tamoxifen treatment severely diminished the performance of cDKO:Cre+ mice without affecting their cDKO:Cre- littermates (**Figure 9A**). Interestingly, the deterioration of scotopic vision was somewhat resistant to tamoxifen treatment until day 22. These observations are in a good agreement with the data from the water maze task (**Figure 9B**). Thus, the loss of RGS affects mouse scotopic vision in a stereotyped manner that does not depend on the paradigm chosen to evaluate behavioral performance.

Finally, we used the OKR approach to evaluate the effect of gradual RGS loss on photopic vision. The parallel processing of cone-generated signals by different classes of BCs makes signaling through ON-BC not entirely necessary for photopic vision (e.g., **Figure 8**). Therefore, we first determined the contribution of cone ON-BC to photopic (70 cd/m$^2$) OKR performance using DKO mice lacking ON-BC responses and compared them to control *Rgs11*−/− littermates that show normal ON-BC function. We found that *Rgs11*−/− and DKO mice had virtually identical contrast sensitivity for slow stimuli

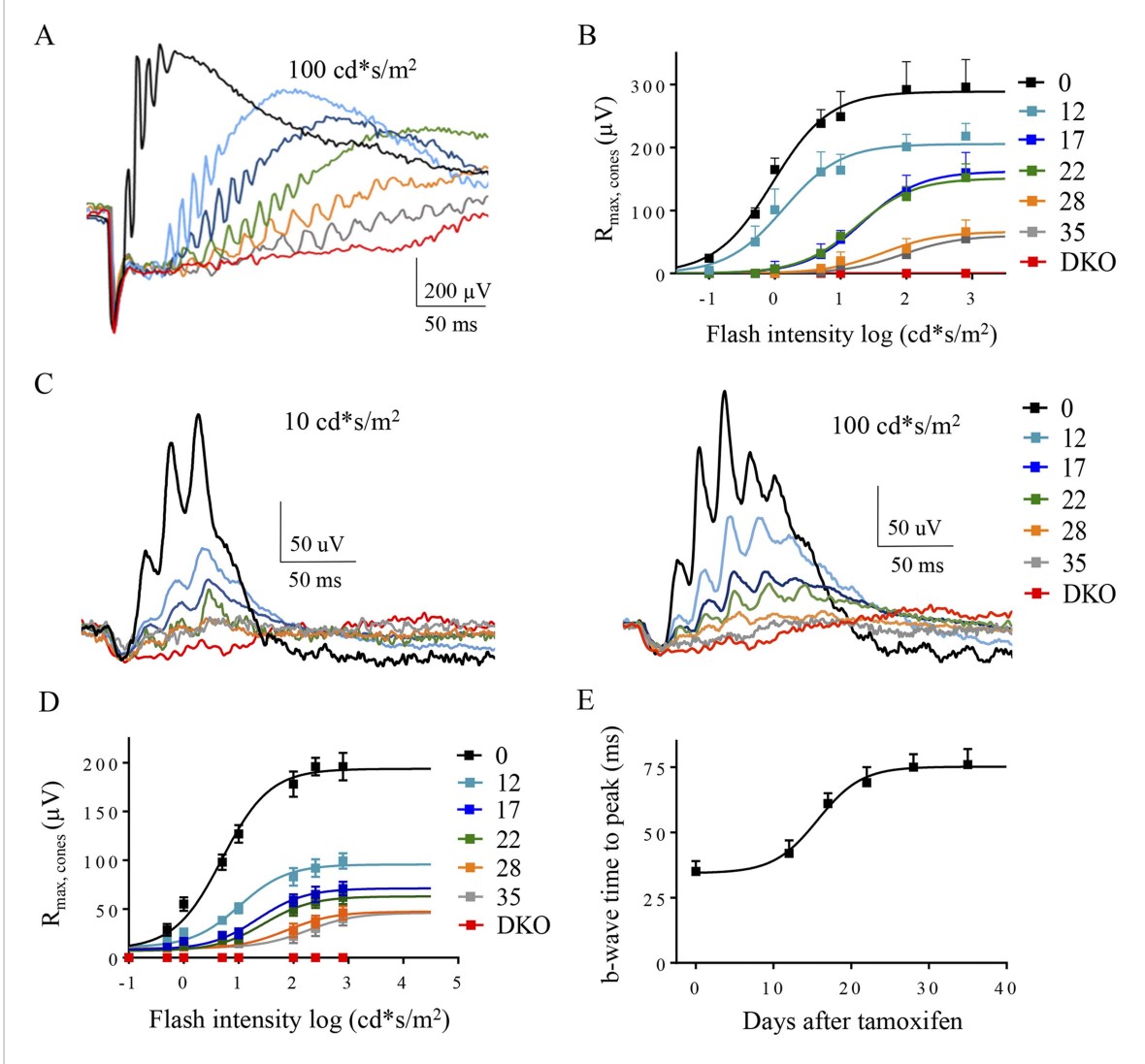

**Figure 7**. The effect of progressive RGS loss on cone ON-BC responses as revealed by photopic ERG. (**A**) Representative dark-adapted ERG responses elicited with photopic (100 cd*s/m²) flashes from cDKO:Cre+ mice at 0, 12, 17, 22, 28, and 35 days post-tamoxifen administration. (**B**) Cone-generated component of the ERG response extracted from fitting the second phase of the response in *Figure 5B*. (**C**) Representative light-adapted ERG responses elicited with photopic flashes (10 cd*s/m², left) and (100 cd*s/m², right). Steady background light of 50 cd/m² was applied to saturate rods. (**D**) ON-BC dose-response plot of maximal ERG b-wave amplitudes recorded with background light and plotted against their eliciting flash intensities. (**E**) Analysis of the changes in the ERG b-wave onset timing elicited by half-saturating flash intensities on a light background. Errors bars are SEM. *Figure 2—figure supplement 1*: isolation of the ERG b-wave peak.

The following figure supplement is available for figure 7:

**Figure supplement 1**. Isolation of the ERG b-wave peak.

(up to 12 deg/s), whereas at higher stimuli speeds (Sp) the DKOs displayed impaired discrimination of low-contrast gratings pattern compared to their *Rgs11*−/− counterparts (*Figure 9C*). In contrast, photopic visual acuity of DKO mice remained unaffected across most Sp and showed only a minor decline at the fastest speed of 50 deg/s (*Figure 9D*).

Having established the contribution of the cone ON-BC pathway to overall contrast discrimination, we evaluated the impact of declining RGS concentration on photopic vision. Tamoxifen administration in cDKO:Cre+ mice caused a 2.4-fold reduction in photopic contrast sensitivity observed at maximal stimuli speed of 50 deg/s in as early as 12 days (*Figure 9E*). The average photopic contrast sensitivity

**Table 4.** Effect of tamoxifen treatment on light-adapted photopic ERG b-wave parameters

| Days post tamoxifen | $I_{0.5}$, photopic (cd*s/m$^2$) | $R_{max}$, photopic | Time to peak (ms) at $I_{0.5}$ |
|---|---|---|---|
| 0 | 5 ± 1 | 204 ± 31 | 35 ± 5 |
| 12 | 10 ± 2 | 111 ± 22 | 42 ± 4 |
| 17 | 23 ± 3 | 70 ± 7 | 63 ± 6 |
| 22 | 65 ± 4 | 65 ± 6 | 69 ± 10 |
| 28 | 190 ± 29 | 55 ± 4 | 73 ± 5 |
| 35 | 260 ± 44 | 53 ± 3 | 74 ± 6 |
| DKO | N.A | 0 | N.A |

Values are mean + SEM, n = 5 mice for all timepoints.

reached its lowest level by day 22, with a nearly a fivefold decline compared with either untreated littermates or a group of cDKO:Cre- controls that also received tamoxifen. This is in stark contrast to scotopic conditions ($\sim 3 \times 10^{-4}$ cd/m$^2$) where contrast sensitivity did not start deteriorating until after day 22 following tamoxifen administration, and by day 35 it had reached a sevenfold decline as compared to cDKO:Cre- control mice (*Figure 9F*).

## Sensitivity and kinetics of rod and cone ON-BC responses and behavioral performance of mice correlate with different RGS levels

The availability of quantitative information regarding RGS levels at any time point in the analysis allowed us to determine how signal transmission to ON-BCs influences visually-guided behavior with the RGS concentration as a common denominator (*Figures 10, 11*). Thus, we compared how both ON-BC ERG response parameters (i.e., b-wave onset, amplitude, and sensitivity) and behavioral

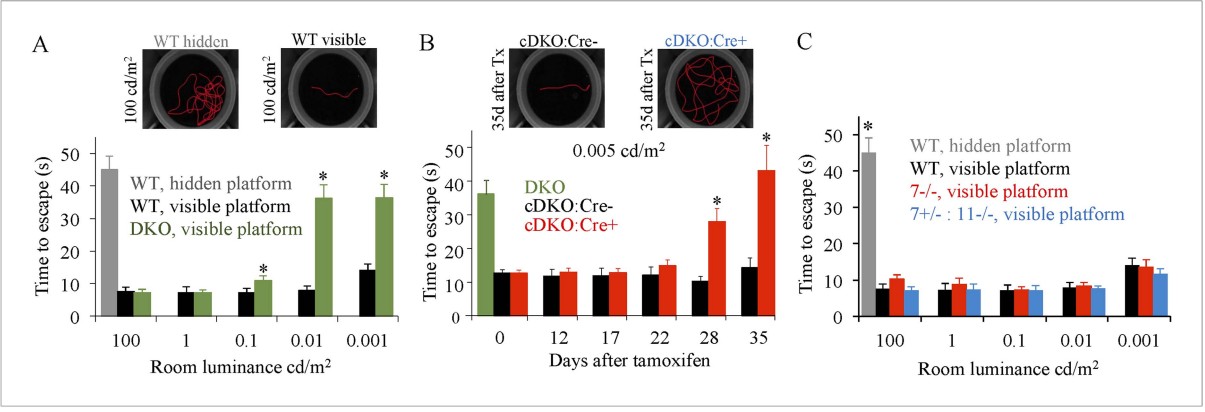

**Figure 8.** Evaluation of scotopic vision using water maze behavioral test. (**A**) Development and validation of visually-guided behavioral task for the assessment of mouse vision. Mice were trained to find a randomly placed visible escape platform in a water maze. Their tracks were recorded and used to determine times to escape from water. Under bright photopic light environment, mice with constitutive elimination of RGS7 and RGS11 (DKO) find visible (7 ± 1 s), but not hidden (45 ± 4 s) platform as readily as wild-type mice (7 ± 1 s). Error bars are SEM (*p < 0.05, t-test, n = 4–5). Escape latencies thus served as a measure of their visual abilities. The room luminance was then decreased in a step-wise fashion followed by the re-assessment of mouse performance. DKO animals performed significantly worse than wild-type (WT) mice in the task at luminance levels ≤0.1 cd/m$^2$ (*p < 0.05, t-test, n = 5 for both groups). (**B**) Behavioral performance of the conditional knockout mice (cDKO) mice in the visual task upon induction of RGS7 loss by tamoxifen administration. Mice were assessed under pure scotopic conditions equating to luminance of 0.005 cd/m$^2$. Performances of cDKO:Cre+ and cDKO:Cre- littermates were compared across days after tamoxifen treatment. At days 28 and 35 cDKO:Cre+ mice performed comparably to DKO mice, taking 28 ± 4 and 42 ± 7 s (vs 36 ± 4 s for DKO) to find the platform (p > 0.05, One-Way ANOVA, n = 5 for both groups). In comparison, performance of cDKO:Cre+ mice lagged that of cDKO:Cre- littermates at day 28 (28 ± 4 vs 10 ± 2 s) and day 35 (42 ± 7 vs 14 ± 3 s) (*p < 0.05, t-test for each time point, n = 5 for each group). (**C**) Elimination of RGS7 or RGS11 along with RGS7 haploinsufficiency does not affect mouse performance in the visual discrimination water maze task. Behavioral performance of WT, *Rgs7–/–*, and *Rgs7+/–:Rgs11–/–* mice was assessed in the water maze based behavioral task across various luminance levels covering the photopic and scotopic ranges. Performance was compared among all genotypes at each specific luminance level.

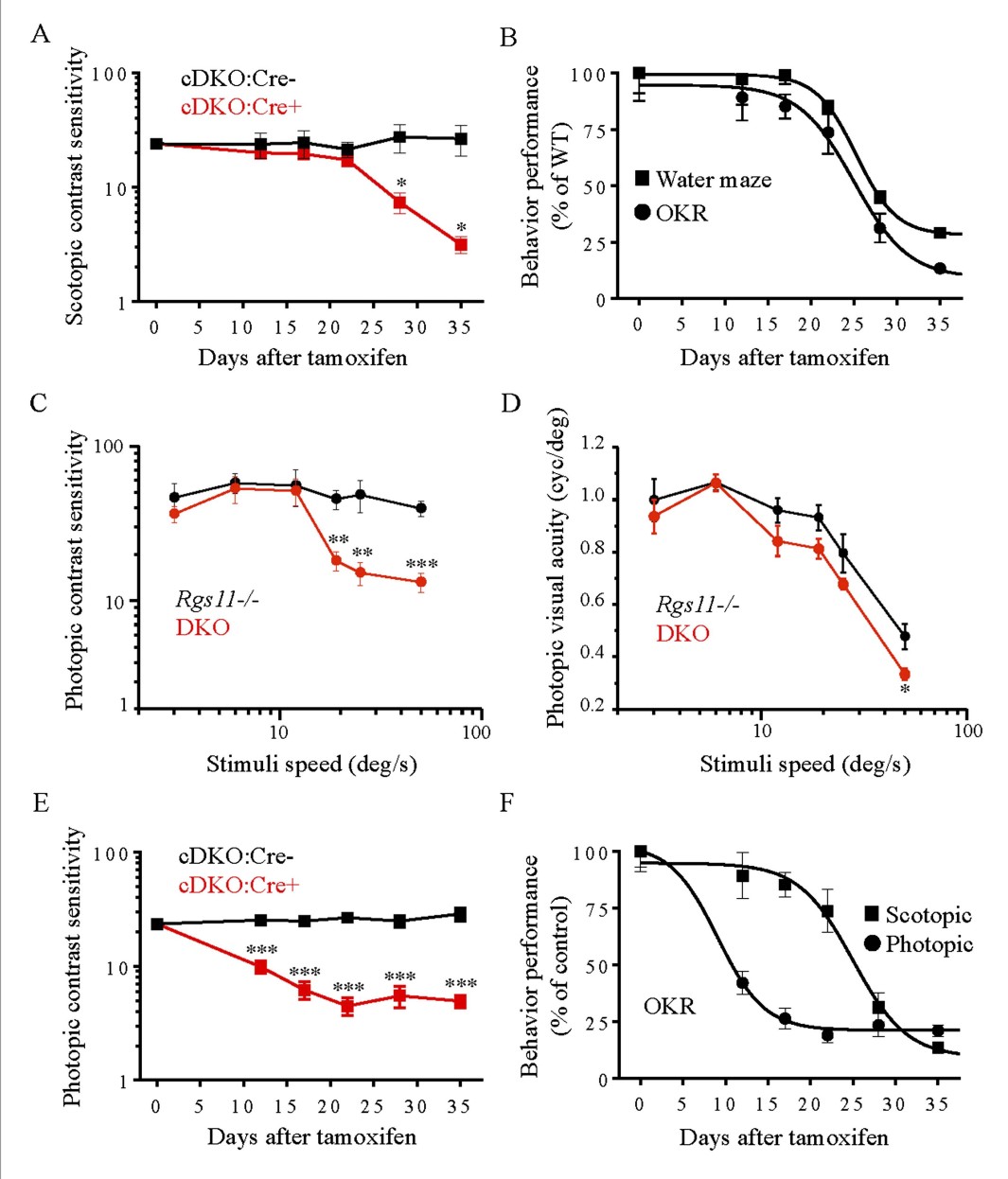

**Figure 9**. Evaluation of scotopic and photopic mouse vision by optomotor task. (**A**) Scotopic (~3 × 10⁻⁴ cd/m²) contrast sensitivity deficits in mice lacking RGS proteins (cDKO:Cre+) vs cDKO:Cre- mice as a function of time of post-tamoxifen administration (*p < 0.05, t-test, n = 4). (**B**) Comparison of visual performance of mice in water maze task and optomotor response tests. Mouse vision declined by half after 25 ± 1 days of tamoxifen administration in both experiments. (**C**) The reliance of high speed contrast sensitivity on intact ON-BC function. Deficits in photopic (70 cd/m²) contrast sensitivity of DKO mice are apparent at stimuli speeds of >12 deg/s (error bars are SEM; t-test: **p < 0.01, ***p < 0.001, n = 4–5). (**D**) Visual acuity is mainly unaffected by the loss of RGS11 together with RGS7 (DKO) as compared to *Rgs11−/−* controls. (error bars are SEM. t-test: *p < 0.05, n = 4–5 mice). (**E**) Photopic (70 cd/m²) contrast sensitivity deficits in mice lacking RGS proteins (cDKO:Cre+) vs cDKO:Cre- controls as a function of time of post-tamoxifen administration (error bars are SEM; t-test: ***p < 0.001, n = 7). (**F**) Comparison of progressive RGS elimination effects on scotopic vs photopic contrast sensitivities of mice. The half-time of contrast sensitivity decline is 25 ± 1 days for scotopic vs 9.0 ± 2 days for photopic conditions, respectively. The data for each group are normalized to their respective values upon beginning of the tamoxifen treatment (day 0) and expressed as percentage.

sensitivity varied as a function of the RGS7 concentration. These results reveal that the effect of RGS7 concentration on b-wave onset kinetics occurred with a half-saturating concentration of 37 ± 2% of available protein (*Figure 10A*). Similarly, a halving of the response amplitude in rod ERG b-wave occurred at RGS7 levels of 39 ± 3% of available protein (*Figure 10B*), and a twofold decline of scotopic response sensitivity occurred at RGS7 levels of 26 ± 2% (*Figure 10C*). A similarly sharp dependence on RGS7 levels was observed for the scotopic vision of mice (*Figure 10D*). Animals

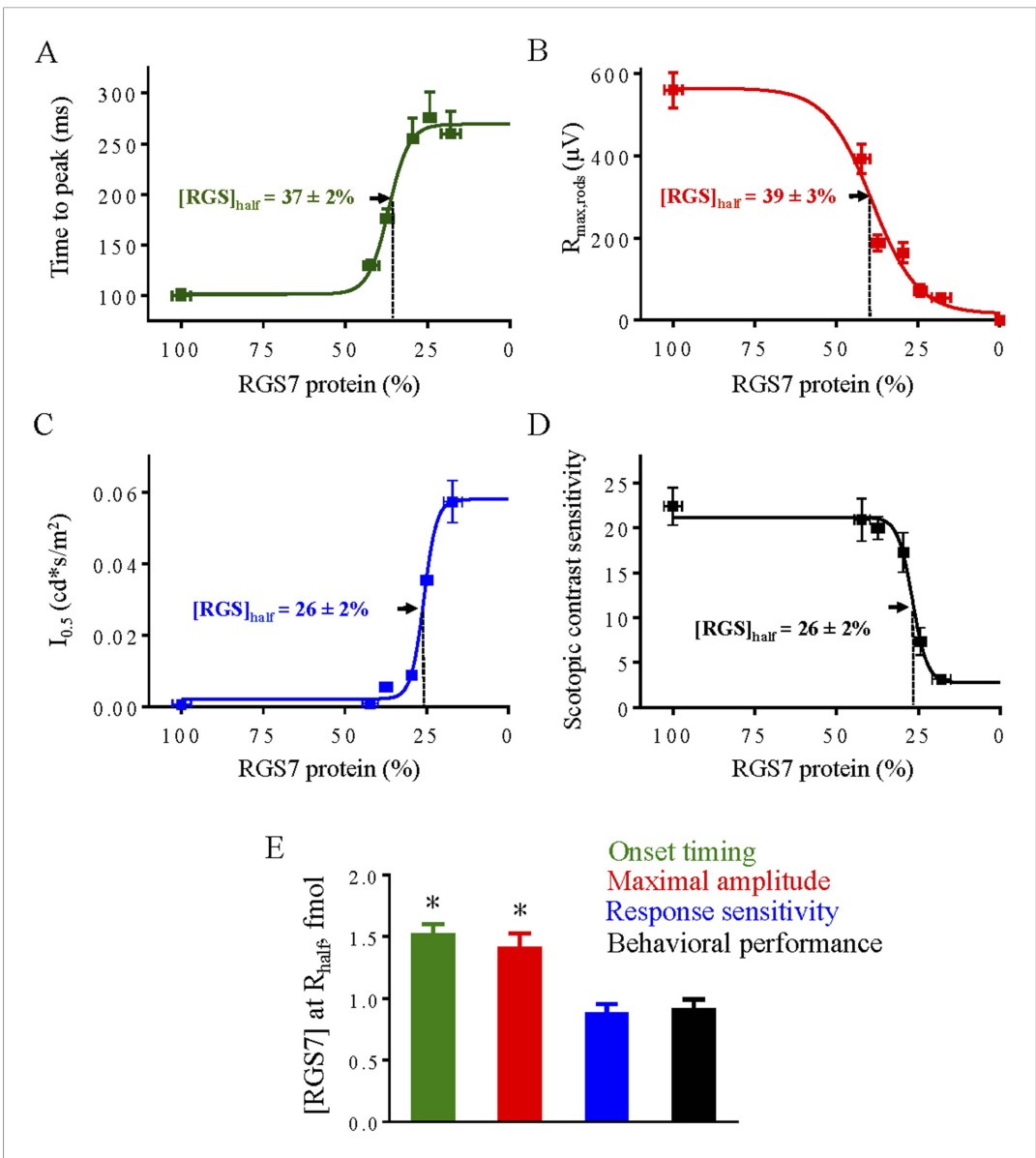

**Figure 10**. Correlation of RGS levels with key parameters of rod-rod ON-BC synaptic transmission and behavioral performance under scotopic conditions. Dose response relationship between half-maximal values of rod ERG b-wave onset timing (**A**), maximal response amplitude (**B**), half-saturating light intensity ($I_{0.5}$) (**C**) and performance in the behavioral task (**D**) were plotted as a function of RGS7 concentration quantitatively determined at each time point following tamoxifen administration. In panels **A**–**D** data points are fitted with sigmoidal dose-response (variable slope) equation with the least squares method. (**E**) Comparison of changes from different response parameters. RGS7% was converted to RGS7 protein levels using quantitative values determined in *Figure 4*. Scotopic behavioral performance is resistant to RGS reduction and tracks most closely with rod ON-BC sensitivity. One-Way ANOVA with Bonferroni's post hoc test reveals that half-maximal values for sensitivity and behavior do not significantly differ from each other ($p > 0.05$, $n = 5$), but differ from those for response kinetics and amplitude ($p < 0.05$, $n = 5$).

showed a precipitous drop in their visual contrast sensitivity once the levels of RGS7 declined below ~35%, with 26 ± 2% reflecting the half-saturating level. To compare directly changes in rod ON-BC response characteristics to behavioral performance, we calculated RGS protein concentration that produced half-saturating responses across these different measures (*Figure 10E*). The data reveals that rod-driven ERG response amplitudes, onset and sensitivity are disproportionately affected by changes in RGS concentration. Although measured response parameters are interrelated by their underlying biochemical mechanism, they correlate differentially with the mouse behavioral performance. For example, a threefold reduction in response amplitude and a twofold deceleration of ERG b-wave onset did not appear to have an appreciable effect on scotopic visual performance. Instead, mouse scotopic behavior tightly correlated with the rod ON-BC response sensitivity, albeit compounded by pronounced changes in other response parameters.

A similar relationship was derived for the cone-driven photopic ERG responses and behavioral sensitivity (*Figure 11*). We found that a halving of the photopic ERG b-wave amplitude occurred at an RGS7 concentration of 43 ± 2% of available protein, a twofold effect on onset kinetics occurred at an RGS7 concentration of 38 ± 2% of available protein, and a twofold decline in sensitivity occurred at an RGS7 concentration of 25 ± 2% of available protein. Thus the photopic ERG b-wave amplitude was the response parameter most sensitive to the reduction in RGS7. However, in contrast to rod-mediated vision, we found that behavioral performance relying on the cone ON pathway was very sensitive to even small decline in RGS7, losing half of their photopic visual performance with 43 ± 2% of the RGS remaining (*Figure 11D*). To compare changes in cone ON-BC response characteristics to behavioral performance we plotted the absolute RGS protein concentration that produced half-saturating responses (*Figure 11E*). We found that photopic vision is more sensitive to changes in the amplitude and kinetics of the cone ON-BC light-evoked responses rather than their light sensitivity.

## Discussion

One of the biggest challenges in neuroscience is to understand how the regulation of cellular activity influences the properties of signals in neuronal circuits to control behavioral responses. These signals may vary in their temporal characteristics, sensitivity thresholds, and throughput bandwidth to ultimately match physiological demands (*Marder, 2012*). GPCR signaling pathways play a key role in setting many circuit parameters as they mediate effects of a vast number of neurotransmitters and neuromodulators (*Nusbaum and Blitz, 2012*). However, our understanding of the molecular mechanisms that shape signaling and their relationship to behavioral outcomes is rather limited.

### Differential modulation of signal transmission in rod and cone ON-BCs

In this study, we used the well-defined rod and cone ON circuits of the retina to examine how changing key parameters of GPCR-mediated responses affected visually-guided behavior. Through genetic manipulations we observed graded alterations in the maximal response amplitudes, timing of response onset, and sensitivity of rod and cone ON-BCs, while testing the effect of these changes on mouse visual performance. We found stark differences between rod- and cone-driven ON circuits with respect to the influence of signal transfer parameters on visually-guided behavior. In the rod circuitry, changes in the timing or amplitude of signal transmission between photoreceptors and ON-BC neurons did not affect behavioral performance in visually-guided tasks that required intact dim vision. However, a change in the sensitivity of transmission at this synapse, albeit aggravated by kinetic deficits, was not tolerated behaviorally and loss of the response sensitivity strictly correlated with the decrease in behavioral performance of the animals. In contrast, cone-mediated vision was sensitive to minor changes in kinetic parameters of signal transmission, causing a pronounced decrease in behavioral performance even when light sensitivity was unaffected. The rod circuit is biased for high sensitivity to detect single photon absorptions close to absolute visual threshold (*Pahlberg and Sampath, 2011*), whereas cone pathways operate at higher light levels and require the capacity to extract temporal features from light stimuli (*Rodieck, 1998*). Thus, our observations suggest that differential specialization of rod and cone pathways involves adaptation at the first synapse in these respective circuits.

Here we find that the parameters of the GPCR-mediated response can be dissociated to provide a hierarchical influence on function. In ON-BCs, decreasing concentrations of two RGS proteins expressed in dendritic tips, RGS7 and RGS11, sequentially influence the onset timing, amplitude, and sensitivity of mGluR6-mediated responses (*Figure 12*). In this pathway, response kinetics and

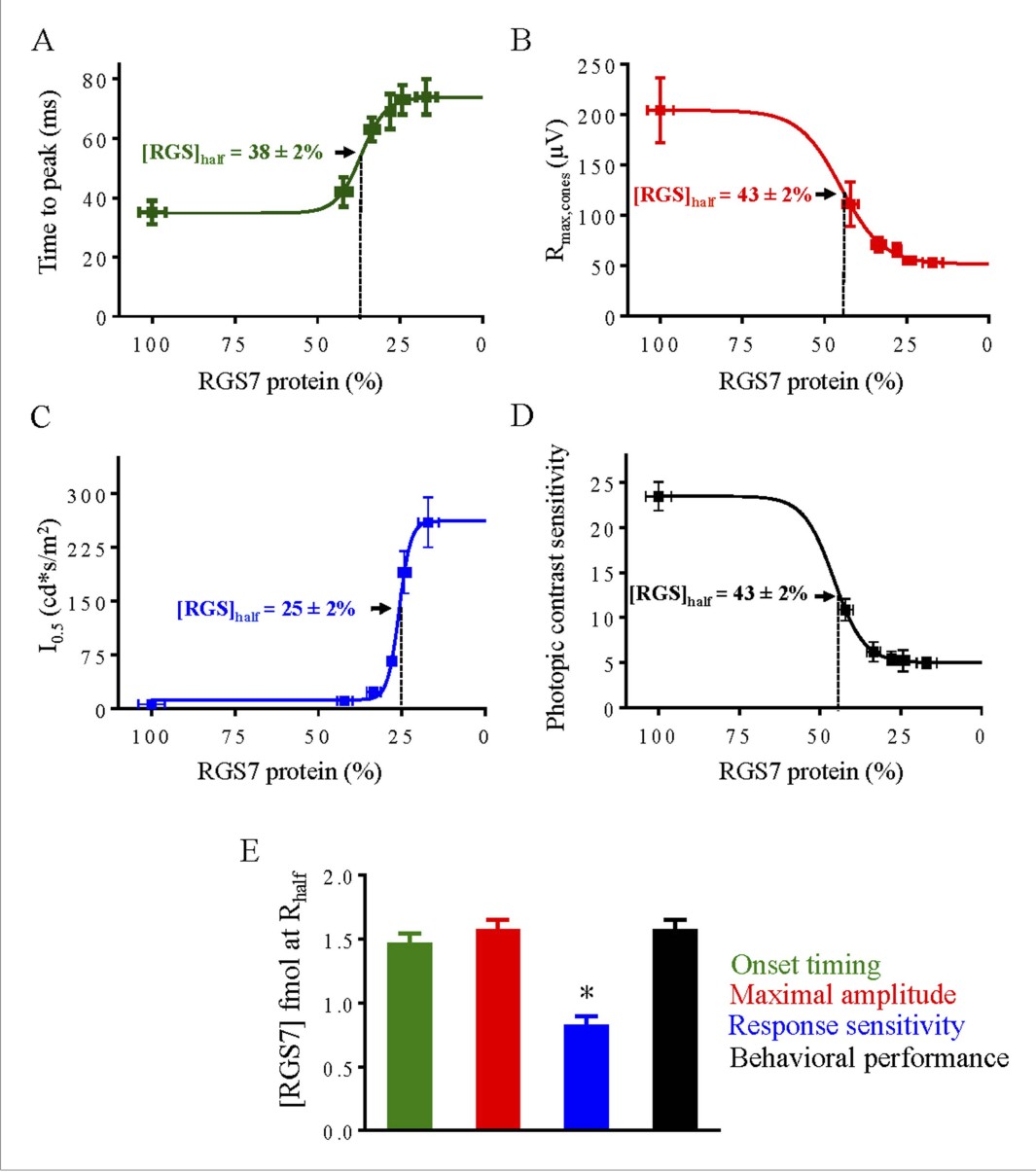

**Figure 11**. Correlation of RGS levels with key parameters of cone-cone ON-BC synaptic transmission and behavioral performance under photopic conditions. Dose response relationship between half-maximal values of cone ERG b-wave onset timing (**A**), maximal response amplitude (**B**), half-saturating light intensity ($I_{0.5}$) (**C**) and performance in the behavioral task (**D**) were plotted as a function of RGS7 concentration quantitatively determined at each time point following tamoxifen administration. In panels **A**–**D** data points are fitted with sigmoidal dose-response (variable slope) equation with the least squares method. (**E**) Comparison of changes from different response parameters. RGS7% was converted to RGS7 protein levels using quantitative values determined in *Figure 4*. Photopic behavioral performance is highly affected by RGS reduction and tracks closely with response amplitude and kinetics. One-Way ANOVA with Bonferroni's post hoc test reveals that half-maximal value for sensitivity differs from all the other parameters ($p < 0.05$, $n = 5$).

amplitude appeared to be the parameters most sensitive to changes in RGS concentration, whereas sensitivity required a much larger reduction in RGS concentration to produce an impact. There is a window in the RGS concentration range where profound effects on kinetics were not accompanied by appreciable changes in response sensitivity. These observations suggest a mechanism by which GPCRs can generate unique patterns of activity and differentially tune circuit properties based on

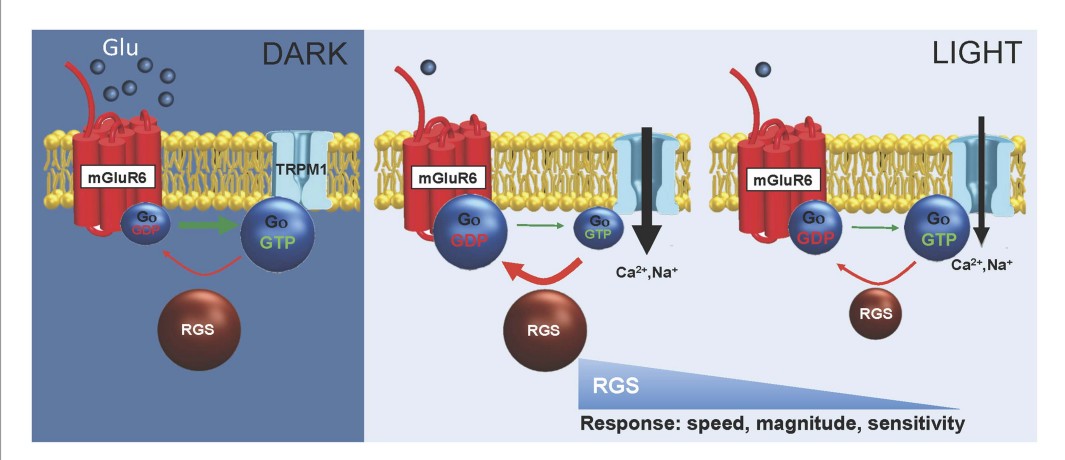

**Figure 12**. Model for RGS function in setting ON-BC responses to light. In the dark, mGluR6 activated by high synaptic glutamate concentration produces ample amounts of GTP-bound Go, which closes TRPM1 channels. RGS proteins (RGS7 and RGS11) deactivate Go converting it back to inactive GDP-bound state but the equilibrium is dominated by excess of Go-GTP ensuring no channel activity. Upon light exposure, the activity of mGluR6 is reduced, and reduction in Go-GTP catalyzed by RGS proteins becomes dominant, allowing channels to open. When concentration of RGS proteins decline below a threshold point the speed of Go deactivation becomes rate-limiting to reduce the speed and magnitude of TRPM1 channel opening as well as its sensitivity to the reduction in glutamate concentration.

varying levels of RGS proteins. Thus, regulatory nodes where information processing is mediated by GPCRs may be tuned to extract temporal features at one end of the RGS concentration range, whereas others may serve as sensitivity filters at the other end of the range. This possibility is supported by observations in photoreceptors where manipulation of RGS9 concentration affects temporal characteristics of the response without changes in sensitivity (*Chen et al., 2000*; *Krispel et al., 2006*) and in hippocampal pyramidal neurons where knockout of RGS proteins changes both response kinetics and sensitivity (*Xie et al., 2010*). Interestingly, levels of multiple RGS proteins have also been shown to fluctuate in response to extracellular cues (*Hurst and Hooks, 2009*; *Traynor et al., 2009*) suggesting that changes in RGS concentration may dynamically control GPCR response properties.

## Saturation of RGS concentration in ON-BCs and reproducibility of signal transfer

Under native conditions, RGS proteins in ON-BCs appear to be present in substantial excess over the minimal amount needed for sustaining wild-type response properties. Rod ON-BCs tolerate an almost 75% reduction in RGS levels before any measurable effects on response properties could be observed. To understand the basis for this overexpression we compared levels of RGS7 and RGS11 to that of the mGluR6 receptors. We found that RGS proteins are present at approximately threefold stoichiometric excess over mGluR6 dimers. Several recent studies suggest that the signal transduction components of the ON-BC are organized in a macromolecular complex (*Morgans et al., 2010*; *Dhingra and Vardi, 2012*). Indeed, both RGS7 and RGS11 form physical complexes with mGluR6 (*Cao et al., 2009*) and a recently discovered orphan receptor GPR179 (*Orlandi et al., 2012*; *Ray et al., 2014*). These interactions are required for proper postsynaptic targeting of RGS7/11 (*Cao et al., 2009*; *Orlandi et al., 2012*). The expression of RGS proteins also depends on mGluR6, and its elimination in mice significantly decreases RGS7 and RGS11 levels (*Cao et al., 2009*). Additionally, mGluR6 and its effector channel TRPM1 are further integrated by a scaffolding protein, nyctalopin (*Cao et al., 2011*; *Pearring et al., 2011*). The relatively rapid (~50–100 ms) response of ON-BCs to flashes of light is unlikely to be supported by the stochastic action of RGS proteins if they were deactivating $G\alpha_o$ in the vicinity of TRPM1 simply by diffusion. Thus, these considerations suggest a model where functionally relevant RGS molecules must be physically integrated into the

macromolecular signaling complex. Since RGS proteins are present in the molar excess over the mGluR6 complex, their depletion does not alter the signaling until the level reaches the stoichiometric threshold and RGS is reduced within the macromolecular assembly. Therefore, it seems reasonable to speculate that overexpression of RGS proteins over the stoichiometry with the mGluR6 signaling complex would minimize fluctuation in the kinetics and sensitivity of ON-BC responses, ensuring their reproducibility. In support of this hypothesis we find that titration of RGS proteins below a certain threshold point (~1:1 stoichiometry with mGluR6 complex) detrimentally affects the signal-to-noise ratio of ON-BC responses and increases their variability.

These results additionally suggest that G protein deactivation normally is not a rate-limiting step in the cascade of events that lead to generation of a depolarizing response in ON-BCs. Previous studies in rod photoreceptors that utilize similar G protein cascade for light reception demonstrated that G protein deactivation limits the recovery of a photoresponse. In these neurons, increase in the concentration of RGS protein (RGS9) resulted in acceleration of the response termination kinetics (*Krispel et al., 2006*; *Burns and Pugh, 2009*). Since a reduction in mGluR6 activity underlies the light-evoked responses in ON-BCs (compared to an increase in rhodopsin's activity for phototransduction), G protein deactivation drives the activation phase of the response and RGS proteins influence its slope. By titrating the RGS concentration in the ON-BCs we found a critical threshold point, below which the onset of the ON-BC depolarization is sensitive to changes in G protein deactivation. The critical threshold point roughly corresponds to ~25% of normal RGS levels (contributed by both RGS7 and RGS11) in wild-type synapses. Notably, a substantial excess in RGS concentration (threefold to fourfold), and hence G protein deactivation rates above this threshold, has no influence on ON-BC response properties. Thus, by setting the levels of RGS expression above this threshold, mGluR6 signaling in ON-BCs ensures that G protein deactivation is not a rate-limiting step in generation of a depolarizing response. Instead, this rate-limiting reaction could be associated with either TRPM1 channel opening or intrinsic deactivation of the mGluR6 receptor. Determining the identity of this rate limiting process that drives the onset of ON-BC depolarizing response will be an important area for future investigation.

## Differential tuning of rod and cone circuits at the first visual synapse

Interestingly, we found that RGS proteins have similar impact on response properties of ON-BCs that form synapses with either rod or cone photoreceptors. Given monosynaptic connectivity between rods and rod ON-BCs, and the progressive night blindness associated with RGS loss, the implications of these changes for rod-mediated behavior are rather unequivocal. However, signaling of cones via parallel ON- and OFF- circuits, together with the heterogeneous nature of cone ON-BCs, complicates the interpretation of the dependence of photopic (cone) vision on cone-to-cone ON-BC signal transmission. The ability to discriminate changes in luminance across a broad range of stimulation frequencies is characteristic feature of photopic vision in many species, including mice (*Umino et al., 2008*). Previous studies have indicated that pharmacological or genetic blockade of the ON-pathway affects contrast sensitivity in mice and monkeys under photopic conditions (*Schiller et al., 1986*; *Iwakabe et al., 1997*). In line with these observations, we find that changes in amplitude and kinetics of the photopic ERG b-wave caused by the loss of RGS expression result in pronounced deficits in the temporal properties of contrast detection for cone vision. Ablation of RGS proteins severely impacted photopic contrast sensitivity for high-speed visual stimuli while leaving visual acuity relatively unaffected. This highlights an essential role that temporal regulation of mGluR6-TRPM1 signaling at the cone-to-cone ON-BC synapse plays for normal photopic vision. Although the details on how RGS proteins may contribute to the responses of individual cone ON-BCs remain to be established, it is possible that differential expression of RGS7 vs RGS11 at cone ON-BC synapses (*Mojumder et al., 2009*) may contribute to setting their unique profiles, thus increasing the temporal resolution of cone vision.

## Materials and methods

### Mouse strains, and tamoxifen gavage

Generation of mice with the constitutive deletion in *Rgs7* (*Rgs7*–/–) (*Cao et al., 2012*), *Rgs11* (*Rgs11*–/–) (*Cao et al., 2008*), or both (DKO) (*Cao et al., 2012*) was previously described. Conditional targeting

of RGS7 was achieved by flanking exon 4 with LoxP sites (*Rgs7flx/flx*) (*Cao et al., 2012*). The resulting mice were crossed with Rgs11−/− line to generate cDKO:Cre- strain (*Rgs11−/−: Rgs7flx/flx*), and further with a Cre driver line ubiquitously expressing tamoxifen-inducible Cre-ERT2 recombinase B6. Cg-Tg(CAG-cre/Esr1*)5Amc/J to produce cDKO:Cre+ mice (*Rgs11−/−: Rgs7flx/flx: CAG-CreERT2*). To induce Cre expression 20 mg/kg tamoxifen (Sigma, St. Louis, MO), dissolved in corn oil and 10% ethanol, was administered by oral gavage with an 18–24 gauge smooth tip needle. Tamoxifen was administered once daily for 5 consecutive days. All procedures were carried out in accordance with the National Institute of Health guidelines and were granted formal approval by the Institutional Animal Care and Use Committees of the Scripps Research Institute (IACUC protocol number 14-001), Washington University (IACUC protocol number 20140236), and the University of Southern California (IACUC protocol number 10890).

## Antibodies, recombinant proteins and Western blotting

The generation of sheep anti-TRPM1 is described (*Cao et al., 2011*). Rabbit anti-RGS7 (7RC1), was a generous gifts from William Simonds (NINDDK/NIH) and the guinea pig anti-mGluR6 antibody was a gift from Dr Takahisa Furukawa (Osaka University). Mouse anti-PKCα (ab11723; Abcam, Cambridge, MA), rabbit anti-RGS7 antibodies (07-237; Upstate Biotechnology, Billerica, MA), mouse anti-CtBP2 (612044; BD Biosciences) and rabbit anti-cone arrestin (ab15282; Millipore, Billerica, MA) were purchased.

Recombinant His-tagged RGS7 and RGS11 were co-expressed in Sf9 insect cells together with Gβ5, via baculovirus-mediated delivery, then purified by Ni-NTA chromatography as described previously (*Martemyanov et al., 2005*). Glutathione S-transferase (GST)-tagged mouse mGluR6 C terminus protein (aa840-871) was expressed in *Escherichia coli* and affinity purified on GSTrap HP column (GE, Fairfield, CT). Protein concentration was determined by bicinchoninic acid (BCA) Protein Assay Kit (Pierce, Carlsbad, CA) and adjusted to reflect the protein purity determined by densitometry of Coomassie blue-stained gels.

Whole retinas were removed from mice and lysed by sonication in ice-cold PBS supplemented with 150 mM NaCl, 1% Triton X-100, and Complete protease inhibitor tablets (Roche, Basel, Switzerland). Lysates were cleared by centrifugation at 20,800×$g$ for 15 min at 4°C. Total protein concentration in the supernatant was measured by using BCA Protein Assay Kit (Pierce, Carlsbad, CA). Supernatants were added with SDS sample buffer (pH 6.8) containing 8 M urea and were subjected to 12.5% SDS/PAGE. Protein bands were transferred onto PVDF membranes, subjected to Western blot analysis first with primary antibodies against RGS7, RGS11 or mGluR6 and then with HRP-conjugated secondary antibodies, and detected by using ECL West Pico system (Pierce, Carlsbad, CA). For the quantitative detection of RGS7 we used rabbit anti-RGS7 antibodies (07-237; Upstate Biotechnology, Billerica, MA). Signals were captured on film and scanned by densitometer, and band intensities were determined by using NIH ImageJ software.

## Immunohistochemistry

Dissected eyecups were fixed for 15 min in 4% paraformaldehyde, cryoprotected with 30% sucrose in PBS for 2 hr at room temperature, and embedded in optimal cutting temperature medium. 12-micrometer frozen sections were obtained and blocked in PT1 (PBS with 0.1% Triton X-100 and 10% donkey serum) for 1 hr, then incubated with primary antibody in PT2 (PBS with 0.1% Triton X-100 and 2% donkey serum) for at least 1 hr. After four washes with PBS with 0.1% Triton, sections were incubated with fluorophore-conjugated secondary antibodies in PT2 for 1 hr. After four washes, sections were mounted in Fluoromount (Sigma, St. Louis, MO). Images were taken with a Leica SP800 confocal microscope. Quantitative analysis of immunofluorescence from confocal images was performed using Leica software. At each time point (day: 0, 12, 17, 22, 28, 35) immunohistochemical staining of cDKO:Cre+ was repeated twice and at least two sections obtained from two individual animals were used for analysis and averaging. Sections were double stained for RGS7 and marker protein mGluR6, which co-localizes with RGS7. Positive staining of mGluR6 was used as a reference for the normalization of the recorded fluorescence intensity in the channel containing RGS7 protein. Rod and cone synapses were differentiated by: (i) counter staining with PNA and/or b-arrestin that specifically label cone terminals, (ii) sublamina position in the outer plexiform layer, (iii) specific clustering pattern of synapses at cone terminals. The fluorescence intensity within synaptic puncta was analyzed using line-scan mode and a constant puncta-encircling area, which tightly surrounded the

contours of each puncta. Mean intensity (measured in pixels) were averaged across ~20 individual punctas per imaged section, taking three sections per retina, and two to three retinas used per time point. Imaging parameters were the same for all sections and retinas. Values for groups were normalized using pre-tamoxifen RGS7 values as a 100% percentage of protein remaining. RGS7 protein percentages were plotted vs the tamoxifen time course and fitted nicely with a simple decay function in GraphPad Prism 6.

## ERG

Electroretinograms were recorded by using the UTAS system and a BigShot Ganzfeld (LKC Technologies, Gaithersburg, MD). Mice (4–8 wk old) were dark-adapted (≥6 hr) or light adapted (50 cd/m$^2$, 5 min) and prepared for recordings by using dim red light. Mice were anesthetized with an i.p. injection of ketamine and xylazine mixture containing 100 and 10 mg/kg, respectively. Recordings were obtained from the right eye only, and the pupil was dilated with 2.5% phenylephrine hydrochloride (Bausch & Lomb, Bridgewater, NJ), followed by the application of 0.5% methylcellulose. Recordings were performed with a gold loop electrode supplemented with contact lenses to keep the eyes immersed in solution. The reference electrode was a stainless steel needle electrode placed subcutaneously in the neck area. The mouse body temperature was maintained at 37°C by using a heating pad controlled by ATC 1000 temperature controller (World Precision Instruments, Sarasota, FL). ERG signals were sampled at 1 kHz and recorded with 0.3 Hz low-frequency and 300 Hz high-frequency cut-offs.

Full field white flashes were produced by a set of LEDs (duration < 5 ms) for flash strengths ≤2.5 cd·s/m$^2$ or by a Xenon light source for flashes > 2.5 cd·s/m$^2$ (flash duration < 5 ms). ERG responses were elicited by a series of flashes ranging from $2.5 \times 10^{-4}$ to $2.5 \times 10^1$ cd·s/m$^2$ in increments of 10-fold. Ten trials were averaged for responses evoked by flashes up to $2.5 \times 10^{-1}$ cd·s/m$^2$, and three trials were averaged for responses evoked by $2.5 \times 10^0$ cd·s/m$^2$ flashes. Single flash responses were recorded for brighter stimuli. To allow for recovery, interval times between single flashes were as follows: 5 s for $2.5 \times 10^{-4}$ to $2.5 \times 10^{-1}$ cd·s/m$^2$ flashes, 30 s for $2.5 \times 10^0$ cd·s/m$^2$ flashes, and 180 s for $2.5 \times 10^1$ cd·s/m$^2$ flashes.

ERG traces were analyzed using the EM LKC Technologies software, Sigma Plot and Microsoft Excel. The b-wave amplitude was calculated from the bottom of the a-wave response to the peak of the b-wave. To isolate the b-wave peak from sharp varying oscillatory potentials, a local regression smoothing algorithm using a bi-square weighting function was used in SigmaPlot (see *Figure 7—figure supplement 1*). Traces obtained from DKO, that completely lack the ON-BC triggered b-wave were subtracted from ERG traces recorded in cDKO:Cre+ mice following tamoxifen administration, each at corresponding light intensities. This was done to remove ambiguity of assigning diminishing b-wave due to interference with P1 component. The data points from the b-wave stimulus–response curves were fitted by *Equation 1* using the least-square fitting method in GraphPad Prism6.

$$R = R_{max,r} \times I/(I + I_{0.5,r}) + R_{max,c} \times I/(I + I_{0.5,c}). \qquad (1)$$

The first term of this equation describes rod-mediated responses (r), and the second term accounts primarily for responses that were cone mediated (usually at flash intensities ≥1 cd·s/m$^2$ for dark-adapted mice; index c). $R_{max,r}$ and $R_{max,c}$ are maximal response amplitudes, and $I_{0.5,r}$ and $I_{0.5,c}$ are the half-maximal flash intensities. Stimulus responses of retina cells increase in proportion to stimulus strength and then saturate, this is appropriately described by the hyperbolic curves of this function. The maximal amplitudes of rod and cone-driven b-wave stimulus–response curves in *Tables 1, 2* are plotted against RGS7 protein concentration and fitted with a simple dose–response curve (variable slope) in GraphPad Prism6. Time to peak of the b-wave was measured as the time from the peak of a-wave to the peak of the b-wave.

## Single cell recordings

Recordings of light-evoked currents in rod ON-BCs were performed as described previously (*Okawa et al., 2010*; *Cao et al., 2012*). Briefly, mice were dark-adapted overnight and euthanized according to protocols approved by the Institutional Animal Care and Use Committee of the

University of Southern California (Protocol 10890). Retinas were isolated, embedded in a low-gelling temperature agar, and 200 μm slices were made on a vibrating microtome (Leica VT-1000S). Retinal slices were placed on the stage of an upright microscope and superfused with heated Ames Media (A-1420; 35-37°C) equilibrated with 5% $CO_2$/95% $O_2$. Whole-cell voltage clamp recordings ($V_m = -60$ mV) were made while delivering flashes of light that varied in strength from evoking a just-measurable response to those generating a maximal response. Noise variance was computed from the 0.2 s preceding the flash. Rod ON-BCs were identified in retinal slices based on the distinctive shape of the cell body and its position in the inner nuclear layer jammed up against the outer plexiform layer. The internal solution for whole-cell recordings consisted of (in mM): 125 K-Aspartate, 10 KCl, 10 HEPES, 5 NMG-HEDTA, 0.5 $CaCl_2$, 1 ATP-Mg, 0.2 GTP-Mg; pH was adjusted to 7.2 with NMG-OH. Membrane currents were filtered at 300 Hz by an 8-pole Bessel filter, and digitized at 1 KHz.

## Evaluation of scotopic vision by behavioral water maze task

Mouse visual behavior was assessed using a water maze task with a visible escape platform. The method for assessing visual function in this experiment is principled on a Morris water maze (*Morris, 1984*) and previous reports describing evaluation of mouse vision by a swimming-based task (*Prusky et al., 2000*). Mice are natural swimmers and this task exploits their innate inclination to escape from water to a solid substrate. This task uses an ability of a mouse to see a visible platform with a timed escape from water, as an index of its visual ability. Before testing, mice readily learned to swim to the visible escape-platform and performance usually plateaued at around 10 s within 15 trials for all treated groups. Mice which did not learn the task, for example, performance did not improve or plateau for at least the last three or more consecutive trials or had any visible motor deficits were discarded from the experiment. Visual-guided behavior was tested at 100, 1, 0.1, 0.01, and 0.001 $cd/m^2$ and timed performances from 20 trials (four sessions of five trials each) for each mouse at each light-intensity were averaged. Uniform room luminance settings were stably achieved by an engineered adjustable light-source and constantly monitored with a luminance meter LS-100 (Konica Minolta, Tokyo, Japan). To be certain that we were measuring the mice's visual ability only and not memory, the platform was placed pseudo-randomly in the water tank and all external visual cues were eliminated.

## Optokinetic evaluation of scotopic and photopic vision

Visual acuity and contrast sensitivity of mice was evaluated from optomotor responses using a two-alternative forced-choice protocol (*Umino et al., 2008*; *Kolesnikov et al., 2011*). Briefly, the animal was placed on a pedestal surrounded by four computer monitors and observed using infrared-sensitive television camera. For scotopic measurements, a round array of six infrared LEDs mounted above the platform was used to visualize the animal. Mice responded to visual stimuli (sine-wave vertical gratings presented by computer using staircase paradigm and invisible to the experimenter), by reflexively rotating their head in either clockwise or counterclockwise direction. The observer's task was to track the direction of optomotor responses and report it to the computer which determined the correctness of the choice (*Prusky et al., 2004*). Photopic visual acuity was determined as the threshold for spatial frequency (Fs) of the stimuli at 100% contrast. Contrast sensitivity was defined as the inverse of obtained contrast threshold values. Photopic responses of constitutive RGS7/11 DKO mice were measured at un-attenuated luminance of monitors (70 $cd/m^2$ at the mouse eye level), over a range of various Sp, from 3 to 50 deg/s. Fs of stimuli was kept constant at its optimal value of 0.128 cyc/deg for all speed regimes thus providing a range of corresponding temporal frequencies (Ft) from 0.38 to 6.4 Hz, according to following equation: $Ft = Sp \times Fs$ (*Umino et al., 2008*). Photopic contrast sensitivities of cDKO mice were determined at specified low (6 deg/s) or high (50 deg/s) Sp at 70 $cd/m^2$ luminance level. Scotopic contrast sensitivity of cDKO mice was evaluated at optimal values of Ft (0.75 Hz) and Sp (0.1 cyc/deg), and a corresponding stimuli speed of 7.5 deg/s. In this mode, the monitor luminance was attenuated to ~$3 \times 10^{-4}$ $cd/m^2$ by several neutral density film filters that formed a cylinder around the animal. All data were analyzed using independent two-tailed Student *t*-test, with accepted significance level of $p < 0.05$.

## Acknowledgements

We thank Mrs Martemyanova for help with animal breeding and genotyping and Dr Alicia Faruzzi Brantley for help with setting up behavioral experiments. This work was supported by NIH grants EY018139 (KAM), DA026405 (KAM), EY017606 (APS), EY019312 and EY021126 (VJK), EY002687 to the Department of Ophthalmology and Visual Sciences at Washington University, and EY000331 to the Jules Stein Eye Institute at UCLA.

## Additional information

### Funding

| Funder | Grant reference | Author |
| --- | --- | --- |
| National Institutes of Health (NIH) | EY018139 | Kirill A Martemyanov |
| National Institutes of Health (NIH) | DA026405 | Kirill A Martemyanov |
| National Institutes of Health (NIH) | EY017606 | Alapakkam P Sampath |
| National Institutes of Health (NIH) | EY019312 | Vladimir J Kefalov |
| National Institutes of Health (NIH) | EY021126 | Vladimir J Kefalov |
| National Institutes of Health (NIH) | EY002687 | Kirill A Martemyanov |
| National Institutes of Health (NIH) | EY000331 | Kirill A Martemyanov |

The funders had no role in study design, data collection and interpretation, or the decision to submit the work for publication.

### Author contributions

IS, JP, AVK, Acquisition of data, Analysis and interpretation of data, Drafting or revising the article; YC, Acquisition of data, Analysis and interpretation of data; VJK, APS, KAM, Conception and design, Analysis and interpretation of data, Drafting or revising the article

### Author ORCIDs

Johan Pahlberg, http://orcid.org/0000-0002-4038-9693

### Ethics

Animal experimentation: All procedures were carried out in accordance with the National Institute of Health guidelines and were granted formal approval by the Institutional Animal Care and Use Committees of the Scripps Research Institute (IACUC protocol number 14-001), Washington University (IACUC protocol number 20140236), and the University of Southern California (IACUC protocol number 10890).

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
