## [Decision Letter]

[Editors’ note: a previous version of this study was rejected after peer review, but the authors submitted for reconsideration. The first decision letter after peer review is shown below.]

Thank you for choosing to send your work entitled “Sensitivity and kinetics of transmission at the first visual synapse differentially impact visually-guided behavior” for consideration at *eLife*. Your full submission has been evaluated by three outside experts and a member of the board of reviewing editors. The four reviewers are impressed with the high quality of the data, the breadth of the analysis, and the importance of the topic. However, there are methodological issues that would need substantial clarification, re-analysis, and/or additional experimentation for the manuscript to be considered at *eLife*. The *eLife* approach to this situation is somewhat different from the approach used by other journals. At *eLife*, we do not want to give the authors a list of major changes that must be made in order to continue with the review process. Instead, a submitted manuscript must be reasonably close to its final form at the time of submission for the review process to proceed to acceptance. We have therefore decided to reject the manuscript, but we want to provide you with the full feedback from the reviewers. We hope that this feedback will be helpful to you going forward. We also note that if you decide to substantially revise the manuscript in light of the comments and resubmit it as a new submission, we would be happy to consider it.

Reviewer #1:

Although the experimental data presented in this paper appears thorough, there are several problems with the analysis of the data that currently cast doubt on the interpretations.

1) Separation of rod and cone components of ERG b-wave signals. Figure 5 purports to plot the response-intensity dependence of the rod-driven component of the ERG b-wave, under control conditions and after tamoxifen application. However, it is crucial to demonstrate that the amplitudes plotted here are genuinely rod-driven, and unfortunately the raw data are not illustrated. What needs to be shown are sets of actual b-wave responses, across the full range of flash intensities, together with the extraction of the rod-driven and cone-driven components; these responses are not available, and so it is not possible to evaluate the measurements. Likewise, the fits of [Disp-formula equ1] (in the subsection headed “Electroretinography”) to the measured amplitudes are not illustrated; these fits need to be illustrated for two reasons. First, it seems unusual that an exponent of unity (rather than a power law) is being applied. But secondly, the magnitude of the cone-driven component (listed in Table 1) is so large (>600 uV, even larger than the rod-driven component of ∼500 uV) that the extraction of the rod-driven component appears susceptible to the precise fitting procedures; however, the significance of this potential problem cannot be evaluated with seeing the results.

2) Figure 6: Sensitivity (6B), and half-saturation normalized response (6C). In Figure 6, the measurements of sensitivity appear to have been taken from Table 3, where only the responsive cells have been included in the averages. The authors need to deal with the issue of the unresponsive cells, because response amplitude and sensitivity appear to be correlated. Thus, the omission of sensitivity measurements from the unresponsive (and presumably very insensitive) cells would seem to bias the plotted measurements.

The text (in the subsection headed “Progressive loss of *Rgs7* changes the kinetics and sensitivity of light-evoked responses in rod ON-BCs”), and the axis label and caption to Figure 6, give very confusing information about what is plotted. The caption and axis label refer to “Half-saturation (normalized) response” as a percentage of maximal. But my understanding of a half-saturating response is that it is half-maximal (i.e. 50%), so I cannot understand what is being plotted. The text refers to “sensitivity” but I cannot see any measure of sensitivity plotted. There seems to be something important here that the authors are not explaining to the reader.

3) Plots in Figure 9. In each of panels A, C and D in Figure 9, the authors show the extrapolated behavior at very low *Rgs7* concentrations asymptoting to a level similar to that obtained at around 20% concentration. In the absence of measurements at lower concentrations, this appears somewhat implausible. For example, in panel C, why would the half-saturation intensity suddenly stop increasing as the *Rgs7* concentration dropped below 20%? And hence, how reliable is this asymptotic level (which is used for curve fitting)? For the data in panel C, it would seem much more informative to plot sensitivity (rather than insensitivity), and to determine the level of *Rgs7* that halves the sensitivity.

Reviewer #2:

This paper used a conditional knockout approach to study how altering the concentration of RGS proteins, which regulate G-protein inactivation, influences visual processing. The approach is a powerful one and the paper contains several interesting observations. Several issues, however, make interpretation of the data difficult.

Major issues:

Comparison of dependence of photopic and scotopic behavioral sensitivity on RGS concentration. The apparent difference in the dependence of photopic and scotopic behavioral sensitivity on RGS concentration is a particularly interesting result of the paper, and it is highlighted as such in the Discussion. Interpretation of this result is difficult, however, since the behavioral task itself differed. How can you rule out that the difference reflects the different task rather than the difference in light level and which retinal circuits are active? Specifically, the concern would be that the sensitivity to changes in the time course of the bipolar responses under photopic conditions reflects the use of a task requiring detection of flickering light inputs, while the task used under scotopic conditions may impose less stringent constraints on temporal sensitivity. This is a critical issue as it undercuts what is probably the most interesting result in the paper. I think it is essential that the same behavioral task be used to evaluate signaling via rod and cone circuits.

Time course of changes in behavior and ERG. The lack of sensitivity of the scotopic behavioral measurements on the ERG amplitude (in the subsection headed “Kinetics and sensitivity of rod ON-BC responses and behavioral performance correlate with different RGS levels”, and Figure 9) is somewhat unexpected. Why would this be the case? Sensitivity is a somewhat artificial measurement from a biological perspective because of the normalization, so it is particular unexpected that behavior tracks it rather than response amplitude. Could initially both signal and noise decline with declining RGS concentration such that behavioral threshold remains fixed? If this is the case, it should be apparent from noise recorded in the rod bipolar cells. An expanded discussion of this issue is needed.

The paper could be written much more clearly. There are numerous sentences that I struggled to interpret. A few examples are:

Abstract: “We demonstrate that key parameters of…”;

Introduction: “Combined with observed modulation…”;

Subsection “Progressive loss of *Rgs7* changes the kinetics and sensitivity of light-evoked responses in rod ON-BCs”: “we first examined retinal responses…”;

Subsection “Saturation of RGS concentration in ON-BC ensures reproducibility of signal transfer”: “Notably, a substantial…”.

There are other examples, and generally the paper needs a careful editing job. There are also some very long paragraphs that could get broken up to help a reader, for example the first paragraph of Results.

Other issues:

The Abstract does not give a good idea of the results in the paper. For example, it does not state that the key manipulation in the paper is to alter the concentration of RGS proteins.

Introduction: Sampath and Rieke is probably not the best reference for the role of Go in On-BC response recovery.

Results, first paragraph, and Figure 2: the similarity of the synaptic connectivity shown in Figure 2 is important as the authors point out. But nothing is done to quantify such similarity. A similar issue arises in the discussion of Figure 3.

The evidence for 1:1:1 stoichiometry (Figure 4) is not particularly strong and not uniquely supported by the data. That part of the text needs to be revised to more accurately reflect the actual data.

Heterogeneity of rod bipolar responses: The ERG data and plots based on it suggest a smooth change in response properties as RGS concentration declines. But the single cell recordings seem to provide a much different picture, with large variability among cells. It would help to document that variability in more detail and explain to a reader why the heterogeneity is not a concern in the comparison of the ERG and behavioral data.

Why do rod bipolar responses change in shape so much? It is not immediately obvious why the dependence of the slope of the rising phase of the response on flash strength should change.

In the subsection headed “Kinetics and sensitivity of rod ON-BC responses and behavioral performance correlate with different RGS levels”: the error bars for the dependence of response properties on RGS concentration seem very small given that they should incorporate error in both determination of RGS concentration and in the response parameter of interest.

Inclusion of recordings from ON cone bipolar cells would enhance the paper. It is not clear why these are not part of the paper at present.

Reviewer #3:

The paper by Sarria et al presents a very nice set of data showing that RGS proteins in the ON bipolar cells are likely present at higher concentration than mGluR6, presumably to ensure that the critical step of deactivating Go is reliable. Interestingly, the effect of reduced RGS on visually driven behavior is not identical in the rod and cone pathways. While in scotopic vision, change in behavior correlates with response sensitivity; in photopic vision, change in behavior correlates with change in kinetics. The paper is well written, but the Abstract is not sufficiently informative.

[Editors’ note: what now follows is the decision letter after the authors submitted for further consideration.]

Thank you for sending your work entitled “Sensitivity and kinetics of transmission at the first visual synapse differentially impact visually-guided behavior” for consideration at *eLife*. Your article has been favorably evaluated by K VijayRaghavan (Senior editor), Jeremy Nathans (Reviewing editor) and four reviewers, including the three who reviewed the first version of the manuscript. One of the reviewers, Noga Vardi, has agreed to share her identity.

I am including the four reviews at the end of this letter, as there are many specific comments in them that will not be repeated in the summary here.

In general, the reviewers were impressed with the importance and novelty of your work. Overall, the experiments appear to be carefully executed. However, all four reviewers find that the data is difficult to interpret in some places, and that the writing needs to be sharpened. This is a complex body of work and clear writing is critical.

We would like to encourage you to resubmit a revised manuscript that addresses the specific issues raised in the reviews below. With respect to the writing, *eLife* is trying to restore some balance to the world of scientific writing. We welcome self-critical comments and we believe that such comments actually enhance the reader's ability to judge the science fairly.

I will offer two examples of scientific writing that I think illustrates the preceding point. They are from two papers on ubiquitin that form the core of the discoveries for which the authors shared the Nobel Prize:

1) Proposed role of ATP in protein breakdown: conjugation of protein with multiple chains of the polypeptide of ATP-dependent proteolysis. Hershko A, Ciechanover A, Heller H, Haas AL, Rose IA. Proc Natl Acad Sci U S A. 1980 Apr;77(4):1783-6.

In the second paragraph of the Discussion section, after summarizing the evidence that leads the authors to propose the conjugation of APF-1 (=ubiquitin) to proteins as a way of marking them for degradation, the authors write: “Evidence that APF-1-proteins are intermediates in the breakdown of denatured protein as proposed in Figure 6 is indirect and inconclusive at this time.”

2) Activation of the heat-stable polypeptide of the ATP-dependent proteolytic system. Ciechanover A, Heller H, Katz-Etzion R, Hershko A. Proc Natl Acad Sci U S A. 1981 Feb;78(2):761-5.

In the last paragraph of the Discussion section, after describing their discovery and characterization of ubiquitin ligase, the authors write: “Evidence suggesting the role of the APF-1-activating enzyme in conjugation and in ATP-dependent protein breakdown is not conclusive at present.”

For beautiful examples of clear writing, let me suggest the following:

a) Photolyzed rhodopsin catalyzes the exchange of GTP for bound GDP in retinal rod outer segments. Fung B, Stryer L. Proc Natl Acad Sci U S A. 1980 May;77(5):2500-4.

b) Flow of information in the light-triggered cyclic nucleotide cascade of vision. Fung BK, Hurley JB, Stryer L. Proc Natl Acad Sci U S A. 1981 Jan;78(1):152-6.

Reviewer #1:

A general issue of very broad interest in biomedical science is how RGS proteins regulate the kinetics of GPCR signaling. A related specific issue of great interest to neuroscience is whether (and if so, how) the properties of synapses where GPCR signaling is present are tuned by RGS proteins.

The first issue was addressed quantitatively in the field of phototransduction, where it was shown that the expression level of *RGS9* controls the limiting rate of deactivation of the GPCR cascade in rod photoreceptors activated by photon capture ([21], Neuron). A useful preparation for investigating the second issue is the synapse between rod and cone photoreceptors with “ON” bipolar cells, as the post-synaptic bipolar cell employs a metabotropic (mGluR6) cascade to control the activation of TRPM1 channels. While the full mGluR6 signaling path in the ON bipolar cells has yet to be determined, there is good published evidence that *Rgs7* and *Rgs11* are both involved (and that the presence of at least one is essential) in the responses of ON-bipolar cells.

This investigation elegantly uses an inducible expression system to conditionally knock down the expression levels of *Rgs7* in *Rgs11*^*-/-*^ in adult mice, and reports highly convincing dependence of the kinetics of the in bipolar cells response on the expression level. The quantification of the knockdown, both with Western analysis and with IHC, is impeccable. The electrophysiological data are beautifully presented and convincing in showing that cone- and rod-signaling through ON-bipolars are differentially affected by the knockdowns. Behavioral data show trends consistent with the physiological results, and the general understanding of conditions that favor rod vs. cone signaling. Overall, the paper beautifully characterizes the dependence of the light responses of rod- and cone- ON-bipolars on the expression level of RGS proteins.

1) In many places paper is not very well written for a general audience, but rather seems addressed to vision experts. Also, in too many places the authors slip into vague broad claims, instead of telling the reader precisely what can be concluded from the results.

2) Figure 1 presents a nice set of control data, but it's very tedious and wasteful of the reader's time, and should be relegated to the supplement with one or two brief sentences in the main text. The figure presentation should start with the current Figure 2, which presents the knockdown strategy and highly convincing evidence for its success at the molecular and cellular level.

3) Figure 4 is very powerful. To help the reader two more panels should be added: (1) a plot of R_max_, rods vs. *Rgs7* level, and of time to peak (scotopic b-wave) vs. *Rgs7* level.

4) Figures 5, 6 and 7. It's very puzzling that the saturating ERG b-wave amplitude is systematically reduced as *Rgs7* expression level declines, while the saturating amplitudes of the individual bipolars is not reduced. Could it be that the loss of *Rgs7* and *11* causes the membrane potential of the bipolars to collapse, so that the driving force on the Trpm1 channels declines in vivo (but provided by the experimenter in single-cell recording)? (What were the zero-current holding potentials?) The authors need to comment on this apparent discrepancy. I think this is a clue to normal RGS function in the cells (see point #7).

5) Figures 8 and 9. The behavioral data provide support for the role of *Rgs7* and *11* in ON-bipolars in regulating vision, but they are not really explained in a manner that a general reader could understand. There from the physiological data, it seems that there are two general effect of the cKO: (1) the loss of ON-bipolar signaling amplitude (as manifest in the b-waves); (2) the slowing of ON-bipolar response kinetics. The authors should attempt to explain to the reader how the cellular losses explain the specific behavioral deficits. To me, most of the defects (except for that in panel F) seem modest, and thus the behavioral data mainly show that normal ON-bipolar signaling is needed for completely normal visual function. However, the results also show that the loss of these signals does not lead to blindness (complete loss of visual function).

6) The paper seems to reach for the stars and miss the moon at times. For example, to me the conclusion of the Abstract—“we demonstrated that key parameters of synaptic transmission: sensitivity, kinetics, and maximal response amplitude have unique ranges of dependence on RGS concentration. We further show that these parameters differentially impact visually guided- behavior mediated by rod and cone ON pathways. These findings illustrate that neuronal circuit properties can account for varying needs by adjusting parameters of synaptic transmission at individually-defined synapses”—is amazingly obtuse. The phrase “a key parameter of synaptic transmission” isn't conventional or even clear. The parameters in question are values of extracted from ERGs or electrical responses of ON-bipolar cells: presumably the actual synaptic signal transmission kinetics—the decrease in glutamate release and consequent mGluR6 and Go de-activation—are not changed. The “unique ranges of dependence” on RGS concentration is never shown (see comment #3 above about how it might be introduced.) The phrase “these parameters differentially impact…” is gobbledy gook. Rather, the loss of *Rgs7* and *11* in cone ON-bipolars apparently results in a more severe phenotype.

7) The paper needs a schematic (say, in the Discussion) that would illustrate how the loss of RGS proteins produces the main ON-bipolar response defects. Even though this would necessarily remain speculative, the general reader needs to be given some understanding of how RGS depletion less to loss of ON-bipolar response amplitude and slowing of the responses. The authors don't ever return to the issue of how their results might impact the understanding of the post-synaptic responses of other neurons known to express RGS proteins.

8) The Introduction could be shortened by at least 30-50% without loss of clarity. The bottom line should be: “… and thus we did the following experiments.”

Reviewer #2:

This “new” submission is essentially a resubmission of the previous manuscript, and I'm afraid I have found it very difficult to evaluate. In some ways the authors have addressed previous criticisms, but on examining the paper more deeply many of the flaws raised previously still seem to be present. For example, it is still problematic to compare the ERG recordings (done with flashes ranging from dim to bright, in darkness or on a single rod-saturating background) with either the behavioral experiments (done at a range of room luminances) or with the OKR measurements (done at visual threshold). The new data in Figure 7 are likewise difficult to interpret, because they seem to be dominated by oscillatory potentials which are not even mentioned; instead the authors appear to take some unspecified measurement within those oscillations as representing the photopic b-wave, because the points plotted in D and E seem very different from what I get looking at the traces in C. Also, it had previously escaped my notice that, while the authors make a big point of the slowing of the ERG b-wave time course in Figure 5, they do not comment on the corresponding results for single-cell recordings; inspection of Figure 6 suggests that the phenomenon may be very different in the rod ON-BCs themselves.

The authors have clearly done a lot of interesting experiments and they have some very valuable data. But what I can't really see is that as a paper it all hangs together in a sufficiently sound and coherent way. As a bottom line, the authors have not actually convinced me of the interpretation in their Abstract that “We demonstrate that key parameters of synaptic transmission… have unique ranges of dependence on RGS concentration”. Nor have they convinced me “that these parameters differentially impact visually guided-behavior mediated by rod and cone ON pathways”.

Reviewer #3:

The paper by Sarria et al. is a nice comprehensive attempt to understand how the quantity of RGS proteins affects parameters of the light response in ON bipolar cells and further influence different behavioral tasks. An emphasis is placed on comparing scotpic to photopic vision. My previous concern about the photopic ERG has been addressed; it seems that the concern about the behavioral assays (that should allow a direct comparison between rod and cone pathways, and between physiology and behavior) has also been addressed.

My major concern is the quantification of mGluR6:*Rgs11*:*Rgs7*. The authors used whole retinas for their quantitative Western blottings; however, mGluR6 and *Rgs11* are present only in ON bipolar cells while *Rgs7* is present also in the IPL. I saw no attempt to correct for that. If *Rgs7* is distributed equally in OPL and IPL (a guess from immunolocalization), then the amount of *Rgs7* will be reduced to about 10 fM and *Rgs7+Rgs11* will equal mGluR6 amount. This quantification is important for understanding the composition of the mGluR6-related macromolecule and the mechanism for the light response. As analyzed here, it appears that RGSs are at excess, but if the IPL is factored in, then each mGluR6 molecule may have only one RGS molecule to modify its cascade. In fact, if the mGluR6 receptor functions as a dimer, then each receptor will have two RGS molecules, a more realistic outcome. This concern however, does not affect the conclusions of the paper regarding the difference between rods and cones (it may affect the Discussion).

Another concern is the quantification of immunostaining in rod bipolar vs ON cone bipolar cells. The authors do not mention how they determined the puncta identity. A common way is to counter-stain with PNA, but this is not demonstrated or mentioned in Methods. I therefore have to conclude that the authors simply used the pattern for this distinction. While the pattern gives some indication about the identity of the puncta, it is far from being adequate for quantification.

Figure 7. It seems to me that the effect of reduced RGS is much greater on R_max_ and I0.5 than on time to peak. The authors should add a graph showing this parameter vs. days after tamoxifen (as they did for scotopic conditions). In particular, we learn later that in photopic conditions, the behavior was mostly affected by amplitude, so it will be nice to see these graphs next to each other.

Reviewer #4:

This paper uses ON bipolar signaling pathways in the retina to correlate the decrease in RGS expression with synaptic transmission and behavior. This is potentially a very powerful approach, and many of the results in the paper are clear and compelling. However, several substantial issues keep the paper from realizing its potential.

Comparison of scotopic and photopic behavior:

One of the central conclusions of the paper is that scotopic and photopic behavior have different sensitivities to properties of bipolar signaling (in the subsection headed “Differential modulation of synaptic transmission parameters in rod and cone ON-BCs”). This is based on Figures 10 and 11, which nicely plot physiological signaling properties and behavior against RGS expression levels. These plots show that scotopic behavior correlates with bipolar sensitivity, while photopic behavior correlates with bipolar kinetics and amplitude. My concern about this analysis is that the behavioral tasks differ. Scotopic behavior was measured using the OKR at speeds corresponding to peak sensitivity. Photopic behavior was measured at higher speeds, since only at those speeds did ON bipolar signaling appear to influence the OKR under photopic conditions. To be convincing, the authors need to show that the particular bipolar feature that behavior correlates with does not depend on task parameters.

Morphology:

A key point in interpreting the results is whether the synapses are altered morphologically or only functionally. The data on this point is not convincing. First, there is little quantification. Second, the resolution of the images provided is not sufficient to evaluate the possibility that synaptic components are rearranged. Third, some key procedures – such as how components at rod and cone synapses were isolated – are not described (or at least I did not find them easily).

Cone bipolar recordings:

Recordings from cone bipolar cells would help interpret the photopic behavioral results and substantially enhance the paper.

Writing:

The writing in many places is sufficiently confusing so as to alter or obscure the conclusions of a sentence. The entire paper needs to be gone through carefully with this issue in mind.

---

## [Author Response]

[Editors’ note: the author responses to the first round of peer review follow.]

The earlier version of this work has been evaluated by *eLife*, and while the paper was formally rejected at the time we were encouraged to resubmit it as new version provided that we address points that required additional work. We are happy to report that we performed extensive additional studies which we hope should clear the concerns raised by the Reviewers. From the summary of discussion among the reviewers there appears to be two major concerns about our study: 1) separation of rod vs cone components measured by ERG (points A and B) and 2) different behavioral paradigms used to evaluate scotopic (rod-dominated) and photopic (cone-dominated) vision (points C and D).

To address the first concern, we performed additional set of experiments where we measured ERG responses on a rod-suppressing light background across all time points following tamoxifen administration. This intensity of this background was very similar to what was used for the behavioral study to attain proper correspondence of the results. We further ensured that truly scotopic light intensities, which do not activate cones, were used for the analysis of rod pathway by ERG. We added this new information, and re-analyzed all the data. In sum, we are convinced that rod and cone driven ERG responses are cleanly separated.

To address the second concern, we established another behavioral paradigm where we evaluated temporal contrast sensitivity under scotopic conditions using optomotor test. Thus, we are now using the same behavioral test to evaluate vision under both scotopic and photopic conditions. Importantly, the results of the optomotor experiments closely matched the results observed in the water maze task, lending further credence to the interpretation that tuning of rod vs cone ON-BC pathways in respect to their reliance on RGS concentration ranges differentially contributes to behavior. Below are our responses to specific reviewers’ comments.

Reviewer #1:

*1) Separation of rod and cone components of ERG b-wave signals.*
Figure 5
*purports to plot the response-intensity dependence of the rod-driven component of the ERG b-wave, under control conditions and after tamoxifen application. However, it is crucial to demonstrate that the amplitudes plotted here are genuinely rod-driven, and unfortunately the raw data are not illustrated. What needs to be shown are sets of actual b-wave responses, across the full range of flash intensities, together with the extraction of the rod-driven and cone-driven components; these responses are not available, and so it is not possible to evaluate the measurements. Likewise, the fits of*
[Disp-formula equ1]
*(in the subsection headed “Electroretinography”) to the measured amplitudes are not illustrated; these fits need to be illustrated for two reasons. First, it seems unusual that an exponent of unity (rather than a power law) is being applied. But secondly, the magnitude of the cone-driven component (listed in*
Table 1*) is so large (>600 uV, even larger than the rod-driven component of ∼500 uV) that the extraction of the rod-driven component appears susceptible to the precise fitting procedures; however, the significance of this potential problem cannot be evaluated with seeing the results.*

We now provide the raw plot of full range of b-wave amplitudes elicited across all light intensities (scotopic to photopic) before extraction of rod vs cone components (Figure 5). We plot the fits according to Equation #1 right on the same graph, which matches experimental data very well. The biphasic curve fit reveals their saturation around predicted intensities for rods and cones respectively. Separately, we show the rod component phase of the response (Figure 5) with respective fits. Furthermore, we provide raw data showing individual ERG traces elicited by scotopic flashes (Figure 5). We would like to note that we are not fitting individual traces (Figure 5) for the analysis but rather extract the peak amplitude values from these traces to be plotted as a function of the flash strength used to elicit them (Figure 5). Values extracted from plots in Figure 5 correspond to Table 2 (conditional KO), not Table 1 (haploinsufficient strains). From the current Figure 5 it can be seen that rod-driven b-wave components are larger than cone-driven components. Furthermore, the rod component clearly reaches saturation, and thus can be fitted precisely. We also caught a mistake that was driving overestimation of cone-driven b-wave amplitudes in Figure 1 and Table 1. This error is now corrected.

*2)*
Figure 6*: Sensitivity (6B), and half-saturation normalized response (6C). In*
Figure 6*, the measurements of sensitivity appear to have been taken from*
Table 3*, where only the responsive cells have been included in the averages. The authors need to deal with the issue of the unresponsive cells, because response amplitude and sensitivity appear to be correlated. Thus, the omission of sensitivity measurements from the unresponsive (and presumably very insensitive) cells would seem to bias the plotted measurements*.

We looked into the issue of unresponsive cells in more detail. After analyzing the data more carefully it became apparent that the fraction of unresponsive cells across the time-course of tamoxifen administration did not substantially change and stayed in the 31% to 46% range.

We thus retracted our earlier claim regarding correlation between tamoxifen administration and increase in the number of unresponsive cells. The appearance of these cells is likely related to stochastic hyperactivation of Cre-recombinase and resultant complete loss of RGS proteins that renders cells completely unresponsive. Given that no parameters are extracted from these cells and that their fraction remains relatively constant throughout the experiment, we think that they are not significantly biasing our sensitivity measurements. Consistent with this, we found almost perfect correspondence in sensitivity measured by both patch clamp and ERG. We clarified these points in the text.

*The text (in the subsection headed “Progressive loss of* Rgs7 *changes the kinetics and sensitivity of light-evoked responses in rod ON-BCs”), and the axis label and caption to*
Figure 6*, give very confusing information about what is plotted. The caption and axis label refer to* “*Half-saturation (normalized) response*” *as a percentage of maximal. But my understanding of a half-saturating response is that it is half-maximal (i.e. 50%), so I cannot understand what is being plotted. The text refers to* “*sensitivity*” *but I cannot see any measure of sensitivity plotted. There seems to be something important here that the authors are not explaining to the reader*.

We corrected Figure 6 to show half-saturating responses plotted in individual graphs and in their respective units for single cell and ERG responses vs. tamoxifen days.

*3) Plots in*
Figure 9*. In each of panels A, C and D in*
Figure 9*, the authors show the extrapolated behavior at very low* Rgs7 *concentrations asymptoting to a level similar to that obtained at around 20% concentration. In the absence of measurements at lower concentrations, this appears somewhat implausible. For example, in panel C, why would the half-saturation intensity suddenly stop increasing as the* Rgs7 *concentration dropped below 20%? And hence, how reliable is this asymptotic level (which is used for curve fitting)? For the data in panel C, it would seem much more informative to plot sensitivity (rather than insensitivity), and to determine the level of* Rgs7 *that halves the sensitivity*.

We have changed data presentation in these figures (now Figures 10 and 11), which now show sensitivity. We further added more experimental data points (28d and 35d) and the plots now more clearly show changes around asymptotes. The only graph where the last point (18%) shows saturation less clearly is half-saturation plot (Figures 10 and 11). However, even if the exact value of half-saturation intensity for the 0-18% range of RGS is far off where the fit predicts it to be, it would not significantly affect the [RGS] half value which we estimate from this analysis. Nevertheless, formal analysis of curve fitting by F test for shows that the sigmoidal dose response curve fit is better than using exponential decay with R^2^=0.0991. Hence, we believe that our estimates of RGS concentration required to change response parameters and behavior are accurate. We further compared all measured parameters according to level of *Rgs7* that halves the response as the reviewer suggested and present this data in panels E of Figure 10 and Figure 11.

Reviewer #2:

*Comparison of dependence of photopic and scotopic behavioral sensitivity on RGS concentration. The apparent difference in the dependence of photopic and scotopic behavioral sensitivity on RGS concentration is a particularly interesting result of the paper, and it is highlighted as such in the Discussion. Interpretation of this result is difficult, however, since the behavioral task itself differed. How can you rule out that the difference reflects the different task rather than the difference in light level and which retinal circuits are active? Specifically, the concern would be that the sensitivity to changes in the time course of the bipolar responses under photopic conditions reflects the use of a task requiring detection of flickering light inputs, while the task used under scotopic conditions may impose less stringent constraints on temporal sensitivity. This is a critical issue as it undercuts what is probably the most interesting result in the paper. I think it is essential that the same behavioral task be used to evaluate signaling via rod and cone circuits*.

We completely agree with the reviewer’s comment. To deal with this issue we evaluated mouse vision using the scotopic contrast sensitivity test. This is exactly the same paradigm that we used to evaluate photopic vision, therefore we think that the data are now directly comparable. Strikingly, we still detect a very large difference in behavioral sensitivity of mice to the RGS loss under photopic vs. scotopic conditions. Importantly, the results obtained in the scotopic contrast sensitivity task also agree very well with measurements in object recognition task in the water maze. Together, these data further boost confidence in the conclusions from this work.

*Time course of changes in behavior and ERG. The lack of sensitivity of the scotopic behavioral measurements on the ERG amplitude (in the subsection headed “Kinetics and sensitivity of rod ON-BC responses and behavioral performance correlate with different RGS levels”, and*
Figure 9*) is somewhat unexpected. Why would this be the case? Sensitivity is a somewhat artificial measurement from a biological perspective because of the normalization, so it is particular unexpected that behavior tracks it rather than response amplitude. Could initially both signal and noise decline with declining RGS concentration such that behavioral threshold remains fixed? If this is the case, it should be apparent from noise recorded in the rod bipolar cells. An expanded discussion of this issue is needed*.

We think that the reviewer’s intuition here is correct. The traditional definition of sensitivity needs to consider the ability to discriminate light-driven signals from the noise, or a signal-to-noise ratio. While most of the emphasis of the manuscript is on evaluating changes in the light induced changes in signal, we had not examined how elimination of RGS proteins affects noise. In the revised manuscript we separately evaluated changes in noise and this information can now be used to approximate changes in signal-to -noise ratio. While the signal per photon absorbed changes in magnitude (as indicated by their half-maximal flash strength), we detected no change in rod ON-BC noise (Table 3). This indicates that the signal-to-noise ratio is declining with time following tamoxifen administration. The lack of effect on behavioral threshold with declining signal-to-noise ratio may then be a reflection of a limiting signal-to-noise ratio that is superceded by the rod ON-BC signal-to-noise ratio as it declines. A treatment of this point is now included in the Discussion.

*The paper could be written much more clearly. There are numerous sentences that I struggled to interpret. A few examples are*:

*Abstract:* “*We demonstrate that key parameters of…*”*;*

*Introduction*: “*Combined with observed modulation…*”*;*

*Subsection “Progressive loss of* Rgs7 *changes the kinetics and sensitivity of light-evoked responses in rod ON-BCs”:* “*we first examined retinal responses…*”*;*

*Subsection “Saturation of RGS concentration in ON-BC ensures reproducibility of signal transfer”:* “*Notably, a substantial…*”*.*

*There are other examples, and generally the paper needs a careful editing job. There are also some very long paragraphs that could get broken up to help a reader, for example the first paragraph of Results*.

We have edited the manuscript, hopefully improving its readability.

*Other issues*:

*The Abstract does not give a good idea of the results in the paper. For example, it does not state that the key manipulation in the paper is to alter the concentration of RGS proteins*.

We modified the Abstract and it now contains reference to RGS proteins.

*Introduction: Sampath and Rieke is probably not the best reference for the role of Go in On-BC response recovery*.

Sampath and Rieke were the first to show directly the specific role of G protein deactivation in limiting the onset of ON-BC responses. This reference is cited in the context of Go deactivation. We have additionally included reference to Dhingra and Vardi (10) who identified Go to be the main G protein species driving depolarizing response of ON-BC.

*Results, first paragraph, and*
Figure 2*: the similarity of the synaptic connectivity shown in*
Figure 2
*is important as the authors point out. But nothing is done to quantify such similarity. A similar issue arises in the discussion of*
Figure 3.

We are unsure about the type of the quantification that might be required here. We see perfect apposition of pre-synaptic (e.g. CtBP2) and post-synaptic (e.g. mGluR6) markers with no difference between genotypes. In the event this alignment is disrupted (as seen in several nob mutants) the synaptic morphology is usually drastically affected and/or either pre- or post-synaptic markers do not reveal staining in the synaptic layer. We have previously examined the morphology of photoreceptor to ON-BC synapses ([5] PNAS) in mice completely lacking both RGS proteins by EM while quantifying structural features and found that it was not affected by RGS loss.

*The evidence for 1:1:1 stoichiometry (*Figure 4*) is not particularly strong and not uniquely supported by the data. That part of the text needs to be revised to more accurately reflect the actual data*.

We revised the manuscript and are now providing exact values for RGS and mGluR6 content measured in our experiments. We would like to note that these experiments were carefully designed and included purified recombinant protein standards spiked into respective mouse knockout retinas. We conducted three independent experiments and generally believe that the data are reliable. We also modified the text to provide more accurate approximation of RGS:mGluR6 stoichiometry.

*Heterogeneity of rod bipolar responses: The ERG data and plots based on it suggest a smooth change in response properties as RGS concentration declines. But the single cell recordings seem to provide a much different picture, with large variability among cells. It would help to document that variability in more detail and explain to a reader why the heterogeneity is not a concern in the comparison of the ERG and behavioral data. Why do rod bipolar responses change in shape so much? It is not immediately obvious why the dependence of the slope of the rising phase of the response on flash strength should change*.

By its nature, ERG recordings reflect average behavior of thousands of cells. It is common to see some variability in a single cell electrophysiological recordings directly from individual bipolar cells. One of the major conclusions of the paper is that in ON-BC RGS are present in substantial excess and this in part ensures reproducibility of their responses and high sensitivity. Thus, it is quite expected that manipulation with concentration beyond the point where it starts affecting the response properties, produces greater variability. This situation is further compounded by the variability associated with the activation of Cre-recombinase which also varies on cell-by-cell basis. Nevertheless, despite the higher than normal variability we were able to perform quantitative analysis of changes in response sensitivity measured in the single cell patch clamp experiments with a reasonable precision (Figure 6). Importantly, we directly compare single cell data and ERG and find them to be in close agreement (Figure 6). We thus believe that heterogeneity of ON-BC responses in our experiments does not detrimentally affect our ability to interpret the data. Furthermore, we mostly base our conclusions on parameters measured by ERG, which show much less variability.

Change in the slope of the light response onset upon RGS loss is expected from the deceleration of the G protein inactivation and resultant slowing of the TRPM1 channel opening and documented previously ([5] PNAS). Again, we find changes in the response onset timing to be in agreement between ERG and single cell recordings.

In the subsection headed “Kinetics and sensitivity of rod ON-BC responses and behavioral performance correlate with different RGS levels”: the error bars for the dependence of response properties on RGS concentration seem very small given that they should incorporate error in both determination of RGS concentration and in the response parameter of interest.

In the revised figures (Figure 10 and Figure 11) we have included error bars that reflect variability in the determination of RGS concentration (X axis) in addition to errors in calculated parameter of interest (Y axis).

*Inclusion of recordings from ON cone bipolar cells would enhance the paper. It is not clear why these are not part of the paper at present*.

Cone ON-BCs are rare when sampling randomly from a slice for patch recordings. Although we agree that having single cell data would have been ideal, obtaining a sufficient data set for the analysis of tamoxifen inducible changes in this rare neuronal population was not feasible in the set of experiments presented. However, we do evaluate the average cone-driven responses of cone ON-BC by ERGs and compare their properties to rod-driven ON-BC responses, with little difference in respect to how parameters of interest in these two neuronal populations respond to change in RGS concentration.

Reviewer #3:

*The paper is well written, but the Abstract is not sufficiently informative*.

We have revised the Abstract to make it more informative.

*[Editors' note: the author responses to the re-review follow*.*]*

Reviewer #1:

1) In many places paper is not very well written for a general audience, but rather seems addressed to vision experts. Also, in too many places the authors slip into vague broad claims, instead of telling the reader precisely what can be concluded from the results.

We have edited the paper substantially to increase its readability for a general audience. This includes elaborating and clarifying how we reach conclusions and substantiate claims.

*2)*
Figure 1
*presents a nice set of control data, but it's very tedious and wasteful of the reader's time, and should be relegated to the supplement with one or two brief sentences in the main text. The figure presentation should start with the current*
Figure 2*, which presents the knockdown strategy and highly convincing evidence for its success at the molecular and cellular level.*

We understand the reviewer’s point and agree that moving current Figure 1 to the supplement would have probably been a better strategy. However, *eLife* policies state that every piece of supplemental material should be related to a particular main figure. In the case of our Figure 1, the material presented provides stand-alone body of data with no logical connection to other figures. In fact, our first submission to *eLife* had this data in the supplement but we were asked to make it a main figure. The figure represents an important pretext to the current work, the fact that knocking down RGS expression to 25% of normal has little effect on the light response properties.

*3)*
Figure 4
*is very powerful. To help the reader two more panels should be added: (1) a plot of R*_*max*_*, rods vs.* Rgs7 *level, and of time to peak (scotopic b-wave) vs. Rgs7 level.*

These analyses are provided in Figure 10. Our concern is that it might be premature to discuss the relationship between RGS levels and their influence on ERG parameters before we present ERG data and walk readers through our analysis of the response parameters.

*4)*
Figures 5, 6 and 7*. It's very puzzling that the saturating ERG b-wave amplitude is systematically reduced as* Rgs7 *expression level declines, while the saturating amplitudes of the individual bipolars is not reduced. Could it be that the loss of* Rgs7 *and* 11 *causes the membrane potential of the bipolars to collapse, so that the driving force on the Trpm1 channels declines in vivo (but provided by the experimenter in single-cell recording)? (What were the zero-current holding potentials?) The authors need to comment on this apparent discrepancy. I think this is a clue to normal RGS function in the cells (see point #7).*

We found that response amplitudes of rod ON-BC are reduced when measured by either ERG or single cells recordings with very good correspondence between these very different methods. The Y scale in Figure 6 now has been set to the same range for every family to show more effectively the reduction in the amplitude of single cell responses and this decline is further documented by the statistical analysis of responses from multiple cells presented in Table 3.

*5)*
Figures 8 and 9*. The behavioral data provide support for the role of* Rgs7 *and* 11 *in ON-bipolars in regulating vision, but they are not really explained in a manner that a general reader could understand. There from the physiological data, it seems that there are two general effect of the cKO: (1) the loss of ON-bipolar signaling amplitude (as manifest in the b-waves); (2) the slowing of ON-bipolar response kinetics. The authors should attempt to explain to the reader how the cellular losses explain the specific behavioral deficits. To me, most of the defects (except for that in panel F) seem modest, and thus the behavioral data mainly show that normal ON-bipolar signaling is needed for completely normal visual function. However, the results also show that the loss of these signals does not lead to blindness (complete loss of visual function).*

We revised the manuscript to provide a more detailed explanation of the behavioral data and their interpretation. The behavioral deficits that we measure for the rod pathway is straightforward: complete loss of scotopic vision. The phenotype is not modest by any means and relationship between the lack of b-wave and night blindness is well established in humans and animal models. “How the cellular losses explain the specific behavioral deficits” is the main investigative line of the paper and we present our thinking about this point in the Discussion, which we have carefully revised and edited for clarity.

*6) The paper seems to reach for the stars and miss the moon at times. For example, to me the conclusion of the Abstract—“we demonstrated that key parameters of synaptic transmission: sensitivity, kinetics, and maximal response amplitude have unique ranges of dependence on RGS concentration. We further show that these parameters differentially impact visually guided- behavior mediated by rod and cone ON pathways. These findings illustrate that neuronal circuit properties can account for varying needs by adjusting parameters of synaptic transmission at individually-defined synapses.*”*—is amazingly obtuse. The phrase* “*a key parameter of synaptic transmission*” *isn't conventional or even clear. The parameters in question are values of extracted from ERGs or electrical responses of ON-bipolar cells: presumably the actual synaptic signal transmission kinetics—the decrease in glutamate release and consequent mGluR6 and Go de-activation—are not changed. The* “*unique ranges of dependence*” *on RGS concentration is never shown (see comment #3 above about how it might be introduced.) The phrase* “*these parameters differentially impact...*” *is gobbledy gook. Rather, the loss of* Rgs7 *and* 11 *in cone ON-bipolars apparently results in a more severe phenotype.*

We have done our best to revise the manuscript and improve clarity. Specifically, we rephrased the statements mentioned by the reviewer hopefully clarifying the points that we were trying to make. The data establishing the dependence of ERG parameters on RGS concentration are presented in Figures 10 and 11. As we mention in our response to comment #3, the analysis that the reviewer is asking for is already reported and serves as a basis for our conclusions. In the revised manuscript, we clarified how we analyzed these changes and used this analysis to inform the conclusions.

7) The paper needs a schematic (say, in the Discussion) that would illustrate how the loss of RGS proteins produces the main ON-bipolar response defects. Even though this would necessarily remain speculative, the general reader needs to be given some understanding of how RGS depletion less to loss of ON-bipolar response amplitude and slowing of the responses. The authors don't ever return to the issue of how their results might impact the understanding of the post-synaptic responses of other neurons known to express RGS proteins.

We have added a schematic to provide our interpretation of this process (Figure 12).

*8) The Introduction could be shortened by at least 30-50% without loss of clarity. The bottom line should be:* “*… and thus we did the following experiments.*”

We revised the manuscript and shortened the Introduction section where we could. However, we felt that dramatic reduction in length might not do the service to a general reader, as it would not allow us to introduce broadly relevant problems at hand.

Reviewer #2:

*For example, it is still problematic to compare the ERG recordings (done with flashes ranging from dim to bright, in darkness or on a single rod-saturating background) with either the behavioral experiments (done at a range of room luminances) or with the OKR measurements (done at visual threshold)*.

All experiments were performed under 2 conditions: scotopic to measure rod pathway and photopic to assess cone pathway function.

For the photopic conditions, we used a rod-suppressing background of 50 cd/m^2^ in ERG while applying 5 to 1000 cd*s/m^2^ flashes with 10 to 260 cd*s/m^2^ half-saturating light intensities used to derive kinetic parameters and 70 cd/m^2^ in photopic OKR making the amount of light delivered to the retinas under different paradigms comparable and representative of the truly photopic conditions.

For the scotopic conditions the flash intensities in ERG ranged from 2.5x10^-4^ to 2.5x10^-1^ to cd*s/m2 and equivalent flash intensities were used for the single cell recordings. Half-saturating light intensities under different tamoxifen treatment time points ranged between 7x10^-4^ and 6x10^-2^ cd*s/m2 and were used for the analysis of kinetic parameters (time to peak). For the behavior, we used 3x10^-4^ (OKR) and 5x10^-3^ (water maze) cd/m2, which are significantly above the visual threshold. The reviewer is correct in pointing out the caveats of comparing electrical responses to behavior under scotopic conditions, mainly because behavioral assessment requires steady light illumination whereas electrophysiological recordings use flashes on a dark background. However, we matched the conditions, to the extent possible, and used alternative behavioral paradigms to evaluate scotopic vision. The light intensities used for the electrophysiological characterization and behavioral evaluation were in the same range corresponding to pure scotopic conditions. Using a mouse model of abrogated rod function (Gnat1-KO) we have previously determined that under those conditions cone-driven responses are negligible and do not impact OKR behavior (20). Furthermore, we found no discernable difference in visual function deterioration when comparing the results of OKR to water maze. Similarly, the results of single cell experiments closely matched data from ERG. Considering these points we believe that we can draw valid conclusions from comparing rod versus cone ON-BC responses and visually–guided behavior mediated by rod and cone circuits.

*The new data in*
Figure 7
*are likewise difficult to interpret, because they seem to be dominated by oscillatory potentials which are not even mentioned; instead the authors appear to take some unspecified measurement within those oscillations as representing the photopic b-wave, because the points plotted in D and E seem very different from what I get looking at the traces in C*.

Oscillatory potentials are a very common feature of photopic ERG. Several approaches have been developed to deal with them when analyzing b-wave parameters. Common ways include either frequency filtering the data or applying local smoothing algorithm to the traces with the least square function to eliminate orderly waveform fluctuations (i.e. see Lyubarsky, 1999, JNeurosci., 442; Robson, 2003, JPhysiol, 509). In our analysis, we have used the smoothing approach for normalizing the data. However, we do show raw data before filtering. In the revised manuscript we are providing a more detailed description of this approach and show the results on the manipulation in Figure 7—figure supplement 1.

*Also, it had previously escaped my notice that, while the authors make a big point of the slowing of the ERG b-wave time course in*
Figure 5*, they do not comment on the corresponding results for single-cell recordings; inspection of*
Figure 6
*suggests that the phenomenon may be very different in the rod ON-BCs themselves*.

In fact, we do see similar slowing in the onset kinetics of the currents recorded from the rod ON-BC by single cell as we do by ERG. Referencing trace peaks in Figure 6 to the “X” axis clearly shows that the time to peak increases about 2-fold over the course of 35 day tamoxifen treatment: from ∼150 to ∼ 300 ms. We did not formally analyze time-to-peak parameter from the single cell recordings due to more than usual variability of the responses that become apparent upon RGS loss. We comment on this in the manuscript.

Reviewer #3:

*My major concern is the quantification of mGluR6:*Rgs11:Rgs7*. The authors used whole retinas for their quantitative Western blottings; however, mGluR6 and* Rgs11 *are present only in ON bipolar cells while* Rgs7 *is present also in the IPL. I saw no attempt to correct for that. If* Rgs7 *is distributed equally in OPL and IPL (a guess from immunolocalization), then the amount of* Rgs7 *will be reduced to about 10 fM and* Rgs7+Rgs11 *will equal mGluR6 amount. This quantification is important for understanding the composition of the mGluR6-related macromolecule and the mechanism for the light response. As analyzed here, it appears that RGSs are at excess, but if the IPL is factored in, then each mGluR6 molecule may have only one RGS molecule to modify its cascade. In fact, if the mGluR6 receptor functions as a dimer, then each receptor will have two RGS molecules, a more realistic outcome. This concern however, does not affect the conclusions of the paper regarding the difference between rods and cones (it may affect the Discussion)*.

Our investigations indicate that *Rgs7* in the retina is specifically present in appreciable quantities only in the OPL. It is true that immunostaining picks up the signal in the IPL, however we found that signal to be completely non-specific as it was also present in DKO retinas lacking *Rgs7*. We now present this data in Figure 4—figure supplement 1. In this study we used KO samples to control for the specificity of the bands that we designate as *Rgs7* by western blotting as well. Therefore, we think that our estimate of absolute *Rgs7* content in the retinas is accurate and reflect primarily the amount present in the ON-BC synapses.

We thank the reviewer for reminding us that mGluR6 exists as a constitutive dimer and functions as a single unit. This consideration, indeed calls for an upward adjustment of our RGS:mGluR6 ratio from ∼ 2:1 to ∼ 3:1. We discuss this calculation and its implications in the revised version of the manuscript.

*Another concern is the quantification of immunostaining in rod bipolar vs ON cone bipolar cells. The authors do not mention how they determined the puncta identity. A common way is to counter-stain with PNA, but this is not demonstrated or mentioned in Methods. I therefore have to conclude that the authors simply used the pattern for this distinction. While the pattern gives some indication about the identity of the puncta, it is far from being adequate for quantification*.

We identify cone synapses by counter-staining with b-arrestin which specifically labels cone terminals. In addition, as reviewer mentioned cone synapses are located in a distinct sublamina of OPL and can be easily identified by a characteristic clustering pattern. We use both hallmarks to distinguish between rod and cone synapses. We clarified this point in the revised manuscript and provided a supplementary figure (Figure 1–figure supplement 1) to illustrate how we distinguish cone synapses.

Figure 7*. It seems to me that the effect of reduced RGS is much greater on R*_*max*_
*and I0.5 than on time to peak. The authors should add a graph showing this parameter vs. days after tamoxifen (as they did for scotopic conditions). In particular, we learn later that in photopic conditions, the behavior was mostly affected by amplitude, so it will be nice to see these graphs next to each other*.

We provide the analysis requested by the reviewer in Figure 10 and Figure 11 where we show how R_max_, I_0.5_ and time-to-peak depend on RGS concentration (days after tamoxifen) side by side.

Reviewer #4:

*Comparison of scotopic and photopic behavior*:

*One of the central conclusions of the paper is that scotopic and photopic behavior have different sensitivities to properties of bipolar signaling (in the subsection headed “Differential modulation of synaptic transmission parameters in rod and cone ON-BCs”). This is based on*
Figures 10 and 11*, which nicely plot physiological signaling properties and behavior against RGS expression levels. These plots show that scotopic behavior correlates with bipolar sensitivity, while photopic behavior correlates with bipolar kinetics and amplitude. My concern about this analysis is that the behavioral tasks differ. Scotopic behavior was measured using the OKR at speeds corresponding to peak sensitivity. Photopic behavior was measured at higher speeds, since only at those speeds did ON bipolar signaling appear to influence the OKR under photopic conditions. To be convincing, the authors need to show that the particular bipolar feature that behavior correlates with does not depend on task parameters*.

Two previous studies addressed the issue of mouse performance in OKR task under scotopic conditions. [52] showed that scotopic contrast sensitivity shows an inverted U dependence on temporal frequency with maximum at ∼0.75 Hz (corresponding to 7.5 deg/s stimuli speed at optimal spatial frequency of ∼0.1 cyc/deg) that sharply declines at speeds exceeding 20 deg/s. In contrast, photopic contrast sensitivity of WT mice is far less reduced at high speeds (as also demonstrated in our Figure 9) at similar optimal spatial frequency. Unfortunately, these characteristic differences in mouse OKR responses in the two light regimens did not allow us to use the same set of parameters in our scotopic and photopic OKR measurements. Considering that scotopic OKR responses become unreliable above 20 deg/s and that we observe deterioration of visual function, we chose to only use optimal parameters for scotopic OKR evaluation. On the other hand, since the constitutive *Rgs7/11* DKO mice did not exhibit any deficits in their photopic OKR behavior at lower stimuli speeds (Figure 9), we restricted our analysis to high speed stimulation where the difference between control and cCKO mice was clearly observed.

A second study by [20] examined scotopic OKR when rod pathway function was severely reduced but not completely eliminated (Gngt1 knockout). In that paper, we found that diminishing rod signaling produces parallel down-ward shift in the behavioral performance across spatial frequencies shifting only light sensitivity. Because loss of synaptic transmission (induced by RGS proteins in the ON-BC) is expected to have similar effect on vision as decrease in rod photosensitivity, we do not expect our cDKO model to be any different in terms of the temporal frequency dependence.

Having said this, we would like to emphasize that we evaluated scotopic vision using 2 different behavioral paradigms: OKR and water-maze and found the results to be in almost perfect correspondence. Thus, we think that differences between the tasks that we use do not influence our interpretations.

*Morphology*:

*A key point in interpreting the results is whether the synapses are altered morphologically or only functionally. The data on this point is not convincing. First, there is little quantification. Second, the resolution of the images provided is not sufficient to evaluate the possibility that synaptic components are rearranged. Third, some key procedures – such as how components at rod and cone synapses were isolated – are not described (or at least I did not find them easily)*.

We have previously shown that complete elimination of *Rgs7* and *Rgs11* even early in development does not affect synaptic morphology and the ability of photoreceptors to make synaptic contacts with ON-BC ([5]; PNAS). To address reviewers’ comment we added higher power image documenting intact TRPM1 synaptic clustering and apposition of pre and post synaptic markers (Figure 2—figure supplement 2). We also performed quantification of histochemical architecture in cDKO retinas post tamoxifen treatment. Finally, we expanded our description of how we identify and quantify rod and cone synapses, providing additional supporting evidence (Figure 2—figure supplement 1).

*Cone bipolar recordings*:

*Recordings from cone bipolar cells would help interpret the photopic behavioral results and substantially enhance the paper*.

Cone ON-BCs are rare when sampling randomly from a slice for patch recordings. Although we agree that having single cell data would have been ideal, obtaining a sufficient data set for the analysis of tamoxifen inducible changes in this rare neuronal population was not feasible in the set of experiments presented (note that the data on OFF cone bipolar cells in Arman and Sampath (2012) required nearly 1.5 years to collect, and this is for one time point). However, we do evaluate the average cone-driven responses of cone ON-BC by ERGs and compare their properties to rod-driven ON-BC responses, with little difference in respect to how parameters of interest in these two neuronal populations respond to change in RGS concentration.

*Writing*:

*The writing in many places is sufficiently confusing so as to alter or obscure the conclusions of a sentence*. *The entire paper needs to be gone through carefully with this issue in mind.*

We appreciate the reviewer’s attention to writing issues. We revised the manuscript editing it for grammar and usage, hopefully making the result more readable.